# Meta-learning Population-based Methods for Reinforcement Learning

**Johannes Hog**                                                          *hogj@cs.uni-freiburg.de*
*University of Freiburg, Germany*

**Raghu Rajan**                                                          *rajanr@cs.uni-freiburg.de*
*University of Freiburg, Germany*

**André Biedenkapp**                                                   *biedenka@cs.uni-freiburg.de*
*University of Freiburg, Germany*

**Noor Awad**                                                            *awad@cs.uni-freiburg.de*
*University of Freiburg, Germany*

**Frank Hutter**                                                             *fh@cs.uni-freiburg.de*
*ELLIS Institute Tübingen, Germany & University of Freiburg, Germany*

**Vu Nguyen**                                                                      *vu@ieee.org*
*Amazon, Australia*

**Reviewed on OpenReview:** *https://openreview.net/forum?id=d9htascfP8*

## Abstract

Reinforcement learning (RL) algorithms are highly sensitive to their hyperparameter settings. Recently, numerous methods have been proposed to dynamically optimize these hyperparameters. One prominent approach is Population-Based Bandits (PB2), which uses time-varying Gaussian processes (GP) to dynamically optimize hyperparameters with a population of parallel agents. Despite its strong overall performance, PB2 experiences slow starts due to the GP initially lacking sufficient information. To mitigate this issue, we propose four different methods that utilize meta-data from various environments. These approaches are novel in that they adapt meta-learning methods to accommodate the time-varying setting. Among these approaches, MultiTaskPB2, which uses meta-learning for the surrogate model, stands out as the most promising approach. It outperforms PB2 and other baselines in both anytime and final performance across two RL environment families. The code to reproduce our results is publicly available at `https://github.com/automl/MetaPB2`.

## 1 Introduction

There have been several breakthroughs in Reinforcement Learning (RL)(Sutton & Barto, 2018) in applications like games (Silver et al., 2016; Berner et al., 2019; Badia et al., 2020) and robotics (Andrychowicz et al., 2020; Kalashnikov et al., 2018; Lee et al., 2020). A crucial component of these breakthroughs is hyperparameter optimization (HPO), as RL algorithms are highly sensitive to the selection of their hyperparameters (Eimer et al., 2023; Islam et al., 2017). While it is a standard practice to employ a learning rate scheduler to dynamically adjust the learning rate in Deep Learning (DL)(Goodfellow et al., 2016), most other hyperparameters are kept fixed during training. This is not a suitable approach for RL where we have non-stationary data distributions that necessitate dynamically adjusting all the hyperparameters, not only the learning rate (Mohan et al., 2023; Parker-Holder et al., 2022). These non-stationary data distributions arise due to the learning process of the RL agent. As the agent's policy is updated, its interactions with the environment change, resulting in the generation of new learning data that differs from previous experiences.

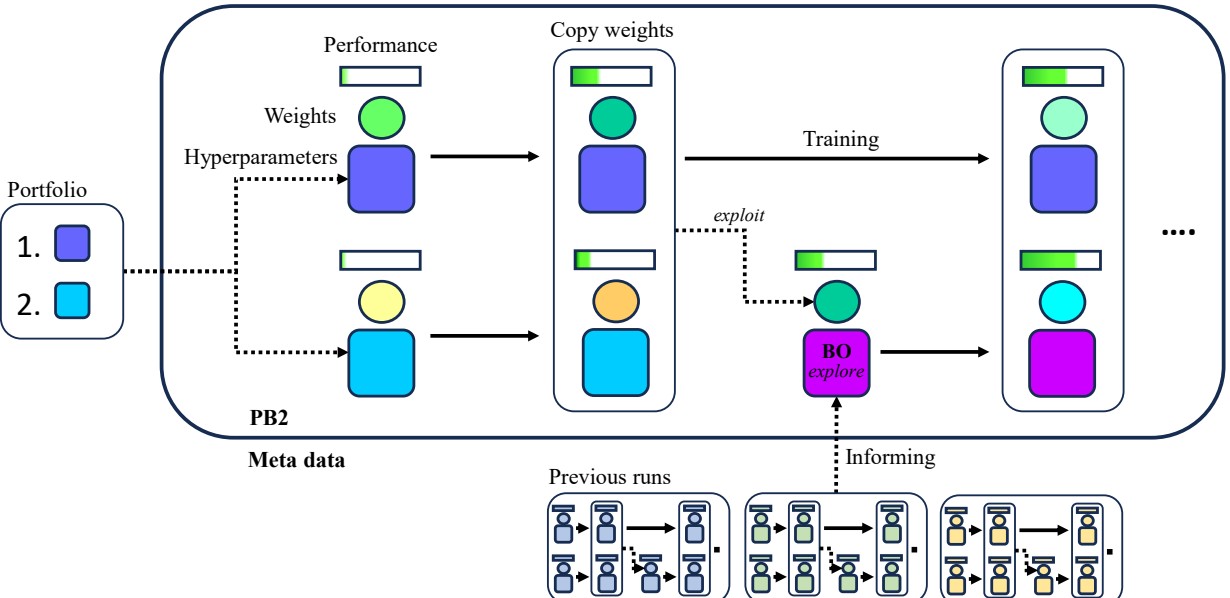

Figure 1: An overview of the different ways we use meta-data to inform PB2. PB2 initializes a population of agents with random hyperparameters and model weights. One of our methods warmstarts the hyperparameters using a portfolio. The population is trained for a fixed number of steps, referred to as the perturbation interval. After this, the weights of well-performing agents are copied over to poorly performing agents (*exploit*) and these agents receive new hyperparameters (*explore*) through Bayesian Optimization (BO). Some of our methods use meta-data data to inform BO. The training cycle then repeats.

One method that dynamically optimizes hyperparameters is PB2 (Parker-Holder et al., 2020a), which optimizes the hyperparameters and trains the agents in parallel, taking only the wall clock time of a single training run (but running on multiple machines). PB2 creates the dynamic hyperparameter schedule by changing the model weights and hyperparameters of poorly performing agents at fixed intervals. The new model weights are copied from a randomly selected well-performing agent, and the hyperparameters are adjusted using Bayesian Optimization (BO)(Garnett, 2023; Snoek et al., 2012) as visualized in Figure 1. A shortcoming of PB2 is that the hyperparameter choices are relatively uninformed during the initial training stages, especially for settings with a small number of examples. We aim to address this by utilizing information from past runs through meta-learning.

We adopted a comprehensive approach to gain an overview of the performance of various methods for the novel application of meta-learning for Bayesian Optimization in a time-varying setting. After discussing background (Section 2) and related work (Section 3), we make the following contributions:

- We propose four different meta-learning methods for the time-varying setting of PB2 (Section 4). In our first approach, which we call TAFPB2, we apply meta-learning to the acquisition function, while for the second approach, we apply meta-learning to the surrogate model using a multi-task GP and call it MultiTaskPB2. In MetaPriorPB2, our third approach, we construct a prior for the GP based on an ensemble from past optimizations. Our fourth approach involves exploring static hyperparameter portfolios to warmstart optimization, aiming to accelerate performance.

- We evaluate and compare the performance of our methods with each other and relevant HPO methods and show that MultiTaskPB2 outperforms the other proposed methods and baselines with respect to anytime and final performance (Section 5.2).

- We investigate a potential synergy of combining our methods by warmstarting TAFPB2, MultiTaskPB2, and MetaPriorPB2. TAFPB2 shows significant improvement, while MetaPriorPB2's

performance degrades. For MultiTaskPB2, warmstarting has no substantial effect on its final performance, where it remains one of the top models. (Section 5.4).

- We investigate the behavior of our methods and perform ablations. First, we qualitatively confirm that our models can identify similar tasks (Section 5.3). Furthermore, we compare runtimes (Section 5.6) and find that the overhead introduced by our methods is large for simpler environments but becomes negligible for larger environments with longer training times. Additionally, we perform an ablation that shows the significant impact of the amount of training the warmstarting portfolio is optimized for. The ablation supports our theoretically motivated default value (Section 5.5).

Among our approaches, MultiTaskPB2, which focuses on the surrogate model, stands out as the most robust and effective among them. It consistently delivers top performance in both anytime and final performance metrics in our experiments and surpasses our baselines.

## 2 Background

Our method builds upon PB2 (Parker-Holder et al., 2020a), which extends Population-Based Training (PBT) (Jaderberg et al., 2017).We provide detailed descriptions of both in this section.

**Population Based Training:** PBT is an evolutionary algorithm that dynamically optimizes hyperparameters by training a population of deep neural network models in parallel. PBT randomly initializes the models and their hyperparameters each of which is independently trained by an asynchronous worker. PBT periodically performs *exploit* and *explore* steps. We dub the period as the *perturbation interval*. In the *exploit* step, the hyperparameters of the top-performing fraction (typically top 20%) of the models are copied over to the poor-performing fraction. Then in the *explore* step, the hyperparameters of these newly copied models are either perturbed or resampled.

**Population-based Bandits (PB2):** PBT can demand large populations for effective performance, particularly due to its exploratory nature. To mitigate this, PB2 extends PBT by integrating Bayesian Optimization (BO)(Garnett, 2023; Snoek et al., 2012) and Gaussian Processes (GP)(Rasmussen & Williams, 2006) for exploration. This adaptation enhances performance efficiency with fewer workers and offers convergence guarantees.

We provide the details on the regular BO and GPs in Appendix A.1 and review the time-varying GP and the selection mechanism used in PB2 in the following.

Using BO in the parallel and time-dependent setting of PBT requires some adjustment of the standard BO components described in Appendix A.1. The observations that make up the dataset $\mathcal{D}$ now additionally contain a time component $t \in \mathbb{N}$ that specifies at what perturbation interval the observation was made, and the performance $\gamma \in \mathbb{R}$ at the start of the last perturbation interval, i.e., $\big((t, \gamma, x), y\big) \in \mathcal{D}$. Furthermore, the targets $y \in \mathbb{R}$ now encode the performance change during the last perturbation interval.[1]

For the GP surrogate model, PB2 models the similarity between input pairs $(t_1, \gamma_1, x_1)$, $(t_2, \gamma_2, x_2)$ with a separable time-varying kernel (Bogunovic et al., 2016)

$$k_{TV}\big((t_1, \gamma_1, x_1), (t_2, \gamma_2, x_2)\big) = k_{Time}(t_1, t_2)k_{SE}\big((\gamma_1, x_1), (\gamma_2, x_2)\big)$$

where $k_{Time}(t_1, t_2) = (1 - \omega)^{\frac{|t_1 - t_2|}{2}}$ is a time kernel with a parameter $\omega \in (0, 1)$ that controls how related observations at different points in time are. It results in a posterior distribution given by

$$\mu(t, \gamma, x) = \mathbf{k_{TV}}(t, \gamma, x)^T(\mathbf{K_{TV}} + \eta\mathbf{I})^{-1}\mathbf{y}$$

$$\sigma^2(t, \gamma, x) = k_{TV}\big((t, \gamma, x), (t, \gamma, x)\big) - \mathbf{k_{TV}}(t, \gamma, x)^T(\mathbf{K_{TV}} + \eta\mathbf{I})^{-1}\mathbf{k_{TV}}(t, \gamma, x),$$

---

[1]$\gamma$ is not part of the original formulation of PB2, but is used in the Ray (Liaw et al., 2018) implementation of PB2, which we use in our experiments. Adding it allows the surrogate model to capture that models with different initial performances see a different performance increase even when having the same hyperparameters.

where $\mathbf{k_{TV}}(t, \gamma, x)$, and $\mathbf{K_{TV}}$ are induced from the time-varying kernel following Eq. (2) from Appendix A.1.

During the *explore* step, PB2 copies the model (including the neural network weights) of a better performing worker at timestep $\bar{t}$ with initial performance $\bar{\gamma}$. Then, PB2 performs BO and selects a new configuration $x^*$ by maximizing GP-UCB (Srinivas et al., 2010)

$$\text{GP-UCB}_{\bar{t},\bar{\gamma}}(x, \mathcal{D}) = \mu(\bar{t}, \bar{\gamma}, x) + \sqrt{\beta}\sigma(\bar{t}, \bar{\gamma}, x),$$

over $x$ while keeping $\bar{t}$ and $\bar{\gamma}$ fixed. The pseudo-code for the PB2 algorithm is provided in Appendix A.2.

## 3   Related Work

Our approach builds on population-based approaches and meta-learning approaches. In this section, we present related work under these two categories.

**Population-based approaches**   draw inspiration from Evolutionary Algorithms (EAs)(Bäck, 1996). PBT (Jaderberg et al., 2017) has emerged as a prominent method, achieving state-of-the-art results in both supervised and reinforcement learning (RL). However, the dynamic and evolving nature of RL landscapes has led to the proliferation of PBT variants tailored specifically for RL. PB2 (Parker-Holder et al., 2020a) integrates Bayesian Optimization (BO) into PBT, enhancing its adaptability and performance. BG-PBT (Wan et al., 2022) extends this concept to include Neural Architecture Search (NAS), enabling simultaneous evolution of architectures and hyperparameters. Other variants like PBT for off-policy RL (Franke et al., 2021) and backtracking-enabled PBT (Zhang et al., 2021) demonstrate improvements in sample efficiency and adaptation to performance fluctuations. Recent developments have focused on addressing the limitations of traditional PBT approaches. Zhang et al. (2021) introduce a PBT variant that allows PBT to *backtrack* to previously better performing configurations because the performance of RL agents can fall drastically during optimization. Bai & Cheng (2024) and Dalibard & Jaderberg (2021) both try to avoid PBT's greediness by judging population members' fitness based on the rate of improvement rather than the absolute performance. Flajolet et al. (2022) show that PBT can even be implemented on a single machine without the need for expensive parallel training on multiple machines. Li et al. (2019) and Bai & Cheng (2024) both propose generalized versions of PBT. Parker-Holder et al. (2020b) optimize all the members of the population at once, aiming to adapt the diversity of the population to improve the reward-diversity trade-off. HOOF (Paul et al., 2019) adapts hyperparameters on the fly for a single policy gradient agent based on an improvement metric for policy gradients to combat PBT's computational expensiveness. These advancements underscore the dynamic evolution and broad applicability of population-based methods in enhancing RL performance across various domains, from algorithmic efficiency to architectural adaptation and beyond. In our approach, we build on top of PB2 because of its efficiency, and because it uses BO. BO builds a model that can be employed for meta-learning in a variety of ways. Therefore, PB2 lends itself better to exploring meta-learning extensions than other model-free population-based approaches.

**Meta-learning approaches**   (Vanschoren, 2019) can be combined with HPO methods to, e.g., transfer good hyperparameter settings across similar problem instances. One prominent research direction in meta-learning is to meta-learn the different components of Bayesian Optimization. Wistuba et al. (2018) propose a transfer acquisition function (TAF) that combines the expected improvement on the target task with improvement estimates from similar meta-tasks. This similarity is computed based on meta-features. Feurer et al. (2022b) propose a method called RGPE to determine this similarity implicitly. We adjust the weighting and acquisition function to our setting and use them in our approaches. Other approaches focus on meta-learning the surrogate model. Of these approaches, the ones that use freeze-thaw BO (Swersky et al., 2014), a form of multi-fidelity optimization, are most closely related to our time-varying setting. While freeze-thaw BO does not dynamically optimize hyperparameters, it builds a surrogate model after partial training to determine which hyperparameter configurations are promising. Quick-Tune (Arango et al., 2024) meta-learns a deep kernel GP (Wilson et al., 2016) based on DyHPO (Wistuba et al., 2022). CMBO (Lee et al., 2024), on the other hand, uses meta-learning to train a Prior Fitted Network (Müller et al., 2022) surrogate model on real data. Both of these surrogate models need to be pretrained on large amounts of meta-data, which is why our surrogate model approach uses a more classical Multi-Task GP (Swersky et al., 2013). In contrast to these

approaches, portfolio-based approaches focus on initializing HPO with a set of configurations that performed well on similar tasks (Feurer et al., 2015a;b). To avoid the computation of meta-features needed to determine similarity, Feurer et al. (2022a) propose to construct a single portfolio that works well across all tasks. This lack of overhead made us build our warmstarting approach on top of Feurer et al. (2022a)'s work. Model-Agnostic Meta-Learning (MAML) (Finn et al., 2017) is a prominent meta-learning technique that differs from the previous approaches by not transferring hyperparameter configurations. Instead, it learns model initializations that can be quickly adapted to new problem instances within a few gradient updates. Banerjee et al. (2023) apply MAML to model-based RL agents that act as principals and incentivize other agents to act in preferred ways.

## 4 Methods

Meta-learning plays a crucial role in enhancing machine learning performance with limited samples. We aim to improve this line of work in PB2 by presenting four meta-learning extensions.

### 4.1 TAFPB2: Transferring Acquisition Function and Weighting

In our first approach, we consider meta-learning via weighting the acquisition function across related tasks. In particular, we incorporate TAF (Wistuba et al., 2018) with a ranking-weighted Gaussian Process ensemble (RGPE) (Feurer et al., 2022b) as the acquisition function for PB2, which we refer to as TAFPB2. Since the original formulation of TAF considers the static setting, we extend it to handle the time-varying setting of PB2 by incorporating the additional inputs given by the interval step $\bar{t}$ and the initial performance $\bar{\gamma}$. Our formulation treats observations similarly to the static case without introducing additional complexity. This results in the development of the time-varying TAF function that we present as follows:

$$\mathrm{TAF}(x,\bar{t},\mathcal{D}_{\tau+1}) = w_{\tau+1}\mathrm{EI}(x,\bar{t},\mathcal{D}_{\tau+1}) + \sum_{i=1}^{\tau} w_i \max\Big(0, \mu_i(\bar{t},\bar{\gamma},x) - \max_{(\hat{t},\hat{\gamma},\hat{x})\in D_{\tau+1}} \mu_i(\hat{t},\hat{\gamma},\hat{x})\Big),$$

where $w_i$ are normalized weights (defined subsequently) for the meta-tasks $i = 1,\ldots,\tau$ and the target task $i = \tau + 1$. We use the EI acquisition function (Močkus, 1974) with the time-varying component, described in Appendix A.3, as the main component for TAF.

We establish a weighting mechanism between a target task and multiple meta-tasks, which we then combine with TAF, as demonstrated in Feurer et al. (2022b). These weights are determined by how accurately surrogate models, derived from the meta-tasks and the target task, characterize the observed configurations. In particular, weights are modeled as the probability that a model fits the target task best based on their ranking loss. This probability is estimated using Monte Carlo estimation with $S \in \mathbb{N}$ bootstrap datasets $D^s$ that are sampled with replacement from $D_{\tau+1}$. The loss of each model $i = 1, 2, \ldots, \tau + 1$ on each bootstrapped dataset $s = 1, 2, \ldots, S$ is denoted by $l_{i,s}$. This results in a weighting

$$w_i = \frac{1}{S}\sum_{s=1}^{S}\left(\frac{\mathbb{1}\left(i \in \arg\min_{i'} l_{i',s}\right)}{\sum_{j=1}^{t}\mathbb{1}\left(j \in \arg\min_{i'} l_{i',s}\right)}\right). \tag{1}$$

The full procedure to compute the weights can be seen in Section A.4.

### 4.2 MultiTaskPB2: Multi-Task Gaussian Process

For our second meta-learning approach, we employ a multi-task Gaussian Process (GP) (Álvarez et al., 2012). This approach establishes a unified model across multiple meta-tasks, enabling observations from different tasks to influence predictions on other tasks based on inferred task similarities.

Our time-varying setting includes multiple meta-tasks, only some of which are relevant to the target task. Therefore, we only select from the top $k$ most similar meta-tasks based on the adjusted RGPE weighting from Eq. (1). In addition, we collect only observations from the meta-tasks that lie within a window around the time step of interest $t$ since they are more relevant to the considered time step. Next, we infer a

task similarity, which determines how strongly different tasks affect each other within the multi-task GP framework.

Using a multi-task GP, we define the dataset $D$ consisting of entries $((m, t, \gamma, x), y)$ where $m \in \{1, 2, \ldots, \tau+1\}$ indicates the task index from which the observation originated. We follow the Linear Model of Coregionalization (LMC) (Journel & Huijbregts, 1976; Bonilla et al., 2007) to define the kernel. This means that our kernel consists of a sum of separable kernels as follows

$$k_{LMC}([m_1, z_1], [m_2, z_2]) = \sum_{i=1}^{\tau+1} k_C^i(m_1, m_2) k_{TV}^i(z_1, z_2)$$

where $z := \{t, \gamma, x\}$ for brevity, the $k_{TV}^i$ are independent kernels with possibly different parameters with coregionalization kernels $k_C^i(m_1, m_2) = c_{m_1, m_2, i}$ where the coregionalization matrices $[c_{m_1, m_2, i}]_{m_1, m_2=1}^{\tau+1}$ are symmetric and positive definite. The entries of these coregionalization matrices are parameterized and estimated with maximum likelihood resulting in the posterior given by

$$\mu(m, z) = \mathbf{k_{LMC}}(m, z)^T (\mathbf{K_{LMC}} + \eta \mathbf{I})^{-1} \mathbf{y}$$
$$\sigma^2(m, z) = k_{LMC}((m, z), (m, z)) - \mathbf{k_{LMC}}(z)^T (\mathbf{K_{LMC}} + \eta \mathbf{I})^{-1} \mathbf{k_{LMC}}(m, z),$$

where $\mathbf{k_{LMC}}(t, \gamma, x)$, and $\mathbf{K_{LMC}}$ are induced from $k_{LMC}$ following Eq. (2). Since we are only interested in predictions on the target task, we fix $m = \tau + 1$ when using it in the acquisition function.

### 4.3 MetaPriorPB2: Meta-Prior to the Gaussian Process

We propose another variant of the meta model using an ensemble of the mean predictions $\mu_i$ of the meta-GP's as the prior mean of the target process $f_{\tau+1}$ (instead of using zero mean prior)

$$f_{\tau+1} \sim \text{GP}(m, k_{TV}) \qquad \text{where} \quad m = \sum_{i=1}^{\tau} w_i \mu_i$$

where the weights $w_i$ are given by the RGPE in Eq. (1) while excluding the target task. In this time-varying setting, the GP posterior distribution (Bogunovic et al., 2016) is parameterized by

$$\mu(z) = \mathbf{k_{TV}}(z)^T (\mathbf{K_{TV}} + \eta \mathbf{I})^{-1} (\mathbf{y} - \mathbf{m}) + m(z)$$
$$\sigma^2(z) = k_{TV}(z, z) - \mathbf{k_{TV}}(z)^T (\mathbf{K_{TV}} + \eta \mathbf{I})^{-1} \mathbf{k_{TV}}(z),$$

where $z = [t, \gamma, x]$ and $\mathbf{m} = [m(z_i)]_{i=1}^{|\mathcal{D}_{\tau+1}|}$. This meta-prior elegantly integrates previous time-varying surrogate models, which are estimated from meta-tasks, according to their relevance for solving the target task.

---

**Algorithm 1** Portfolio Construction

**Input:** Datasets $\mathcal{D}_1, \mathcal{D}_2, \ldots, \mathcal{D}_\tau$ , portfolio size $s \in \mathbb{N}$
**Output:** Portfolio $\mathcal{P}$ of size $s$
1: $\mathcal{C} \leftarrow \emptyset$       ▷ Create portfolio candidates
2: **for** $i = 1, \ldots, \tau$ **do**
3:     $x^* \leftarrow \texttt{hpo}(\mathcal{D}_i)$
4:     $\mathcal{C} \leftarrow \mathcal{C} \cup \{x^*\}$
5: **end for**
6: **for** $i = 1, 2, \ldots, \tau$ **do**     ▷ Create and normalize performance matrix
7:     **for** $x \in \mathcal{C}$ **do**
8:        $p_{x,i} \leftarrow \texttt{evaluate}(x, \mathcal{D}_i)$
9:     **end for**
10:    $p_{:,i} \leftarrow (p_{:,i} - \min p_{:,i}) / (\max p_{:,i} - \min p_{:,i})$
11: **end for**
12: $\mathcal{P} \leftarrow \emptyset$   ▷ Greedily add configurations to portfolio
13: **while** $|\mathcal{P}| < s$ **do**
14:    $x^* \leftarrow \arg\max_{x \in \mathcal{C} \setminus \mathcal{P}} \frac{1}{\tau} \sum_{i=1}^{\tau} \max_{\bar{x} \in \mathcal{P} \cup \{x\}} p_{\bar{x}, i}$
15:    $\mathcal{P} \leftarrow \mathcal{P} \cup \{x^*\}$
16: **end while**

---

### 4.4 Warm-starting the Population with a Portfolio

We consider a more classical, but effective, meta-learning technique known as warm-starting. This approach is inspired by the portfolio construction method used in Auto-sklearn 2.0 (Feurer et al., 2022a).

In Algorithm 1, we create a portfolio of hyperparameter configurations which we use to initialize the training. First, we construct a candidate set $\mathcal{C}$ of possible portfolio members by searching for the optimal hyperparameter configurations on all meta-datasets with a $\texttt{hpo}$ method, such as using random search. Then, we evaluate

each candidate configuration on all datasets to create a performance matrix $[p_{i,j}]_{i,j=1}^{\tau}$ (line 8). Finally, we build the portfolio by iteratively adding the configuration of the candidate set that maximizes the average best performance of the current portfolio if it has been added to it.

Unlike traditional static HPO approaches to portfolio construction, which typically consider final performance after full training, our method constructs portfolios based on scores at the time step $t$ when configurations are first anticipated to change under PB2. Taking the percentage $\lambda$ as the probability that a configuration is changed after a perturbation interval, the probability that an initial configuration changes for the first time after interval $i$ is geometrically distributed with $\mathbb{P}(i) = \lambda(1 - \lambda)^{i-1}$. Therefore, the expected interval when a given initial configuration is changed is the expectation of the geometric distribution

$$\mathbb{E}_{X \sim \text{Geometric}(\lambda)}[X] = \frac{1}{\lambda}.$$

In Section 5.5 we perform an ablation to evaluate the impact of $\lambda$ on algorithm performance.

## 5 Experiments

This section begins with details on our experimental setup, followed by the presentation of our research findings that address the following research questions.

RQ1 How do our methods compare in performance to standard baselines and each other?

RQ2 Do the RGPE weights correctly identify similar tasks?

RQ3 How does the warmstarting method influence the performance of other meta-methods?

RQ4 Does the theoretical recommendation for perturbation interval for portfolio construction hold true empirically?

RQ5 How much runtime overhead is introduced by our methods?

### 5.1 Experimental Setup

We follow the evaluation scheme of the PB2 paper (Parker-Holder et al., 2020a) for our experiments and optimize the hyperparameters of Proximal Policy Optimization (PPO)(Schulman et al., 2017). We now introduce the benchmark environments used for meta-learning and elaborate on our evaluation strategies. Appendix B.1 provides a more detailed overview of our experimental setup.

**Benchmark environments**  We evaluated our approaches on two sets of RL environments, employing CARL (Benjamins et al., 2023) to generate several slightly different versions of each. This approach mimics a blend of comparable and distinct tasks, enabling us to explore the specific tasks our methods effectively learn from. The first set of environments is classic control (Brockman et al., 2016) specifically *mountain_car, cart_pole, pendulum, acrobot*. We used this cheaper environment to set our methods' hyper-hyperparameters and conduct more compute-intensive experiments. The second set of environments is Brax (Freeman et al., 2021) where CARL allowed us to modify 9 environments (see Appendix B.1). In both sets of environments, we manipulated the gravity across versions to reflect that of five different planets.

**Modification to Ray Implementation**  We made minor modifications to the Ray implementation (Liaw et al., 2018) of PB2 and empirically found it to perform better than the original version. This altered version, referred to as PB2*, serves as the foundation for our new approaches. The key modification that had the most significant impact involves how we manage poorly performing workers after the initial training perturbation period. During this phase, we have performance data from various workers but lack the performance differences necessary for meaningful Bayesian Optimization (BO) steps. Ray's default behavior in such cases is to just *exploit*, resulting in duplicate workers with identical weights and hyperparameters. To address this, we additionally sample entirely new hyperparameter configurations. This approach allows us to explore a broader section of the search space and provide better guidance for subsequent BO steps.

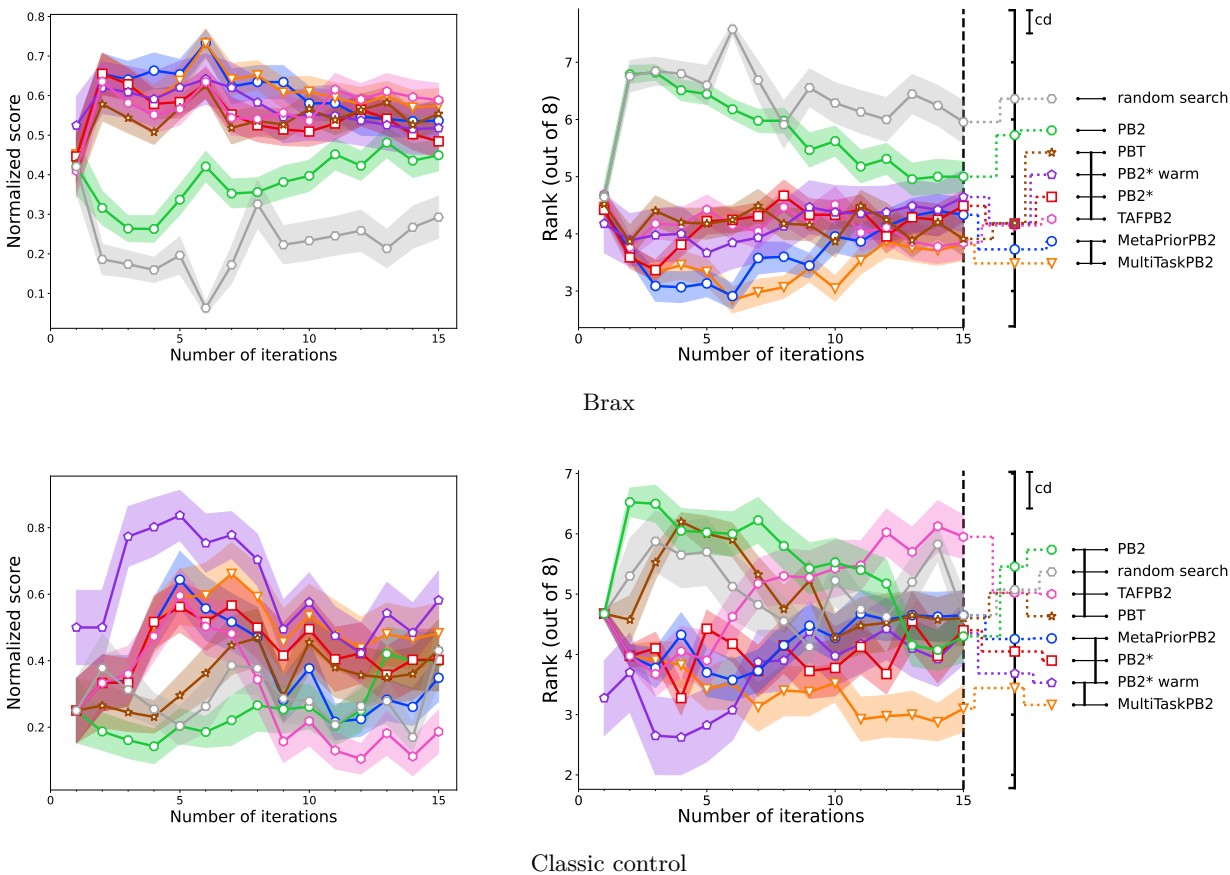

Brax

Classic control

Figure 2: Method comparison with baselines where PPO was trained on Brax (top) and classic control (bottom). Comparing rewards normalized at each iteration (higher is better) on the left and ranking on the right (lower is better). Out of all of our methods and baselines, MultiTaskPB2 consistently performs the best. Note that the different values at the first interval for Brax are caused by crashed experiments (see Appendix B.6). These crashes were caused by divergence and are not fixable without changing the hyperparameter ranges.

**Evaluation** We repeat each of our experiments for 10 seeds and ensure that the initial random configurations are the same for each method given the same seed. Our population-based algorithms split the training into 15 equally sized perturbation intervals. At each of the perturbation intervals, we evaluate every agent for 10 episodes during training. When evaluating a method, we only look at the performance of the best agent at each timestep and compare these in multiple ways. The first way we compare our methods is by looking at normalized rewards. For each environment variant, we normalize the mean rewards of the different methods at each timestep to be between 0 and 1. The second way we compare our methods is by ranking the mean rewards instead of normalizing them. When visualizing the normalized rewards and the rankings, we plot their mean and standard error during the training process. We further pair the rank visualization with a critical difference diagram (Demšar, 2006) using the *Autorank* (Herbold, 2020) Python package to summarize the any-time performance. The diagram displays the mean rank over all timesteps for the methods on a vertical line on the right and vertically connects the lines that indicate the method names if there is no statistically significant difference ($\alpha = 0.05$) between their mean ranks.

### 5.2 RQ1: Methods Comparison

In this section, we address research question RQ1, examining the performance of the proposed methods in comparison to each other and standard baseline methods. Specifically, we compare the novel meta-learning approaches outlined in Sections 4.1 - 4.4 with: PBT (Jaderberg et al., 2017), PB2 (Parker-Holder et al., 2020a), our improved PB2 variant which we dub PB2*, and random search. We use Ray (Liaw et al., 2018) implementations of the baselines since our methods are implemented on top of Ray. The results of these experiments can be seen in Figure 2 while Appendix B.5 shows additional results for different numbers of workers.

Our experiments show that MultiTaskPB2 overall performs the best over both metrics on both environment families. Additionally, our meta-methods demonstrate a statistically significant improvement in anytime performance over PB2. This enhancement in anytime performance appears to primarily stem from the modifications introduced by PB2*. Still, MultiTaskPB2, in particular, was able to use the meta-data to further improve its final performance. In contrast, our other meta-methods do not perform as well. Notably, TAFPB2 exhibits the lowest final performance in classic control environments. While warmstarting enhances initial performance in classic control, it does not substantially impact final performance and shows minimal effect on Brax. Further analysis in Appendix B.5 reveals that a static portfolio fails to provide robust hyperparameter initialization for a wide array of different environments like the ones present in Brax. Individual training plots and raw results can be found in Appendix B.8 and Appendix B.7, respectively.

### 5.3 RQ2: Similarity Weighting

The methods introduced in Sections 4.1 - 4.3, namely TAFPB2, MetaPriorPB2, and MultiTaskPB2, use RGPE weights. These weights measure the similarity between the meta-tasks and the target task. In this section, we explore the capability of RGPE to identify similar environments. We know which environments are similar to each other due to the way they are constructed. Environments with the same base environment but a different gravity should behave more similarly than environments with completely different base environments. For our visualization, we average each meta-task weight for a given target task over all timesteps and seeds and display the results in matrix form.

Since we identify MultiTaskPB2 as the most promising of our approaches in Section 5.2, we will restrict this analysis to MultiTaskPB2. The visualizations for TAFPB2 and MetaPriorPB2 on both Brax and classic control align with the results in this section and can be found in Appendix B.4.

The weight matrix for MultiTaskPB2 on Brax is shown in Figure 3. It has a diagonal block structure with varying block sizes. The smaller blocks are made up of environments that share the same base environment but have different gravity. On average, 34 % of the weight is allocated to them. This is more than three times the 9.5 % of the weight that the average set of environments that uses a different base environment gets allocated. If we look at the base environments of the bigger blocks, we can see that they are related like in the case of *humanoid* and *humanoid_standup* (17 % of the total weight) or *inverted_pendulum* and *inverted_double_pendulum* (20 % of the total weight). In summary, the weights not only identify similarities based on the same base environment but also based on related base environments.

### 5.4 RQ3: Warmstarting the Weighting-based Meta Methods

Our warmstarting approach and other meta-learning methods are orthogonal and can be combined. In this section, we address RQ3 and analyse how this combination affects performance. For this, we trained MultiTaskPB2, TAFPB2, and MetaPriorPB2 with warmstarted hyperparameters and visualize both the normal and warmstarted training in Figure 4.

We can see that warmstarting improves the initial performance in both environment families, though only marginally on Brax. However, MetaPriorPB2 was negatively affected, with its final performance degrading compared to the vanilla version. One explanation for this could be that the meta-prior performs poorly because hyperparameter schedules in the meta-data do not match the schedules we encounter when warmstarting our meta-prior. In contrast to MetaPriorPB2's decreased performance, TAFPB2 shows a significant

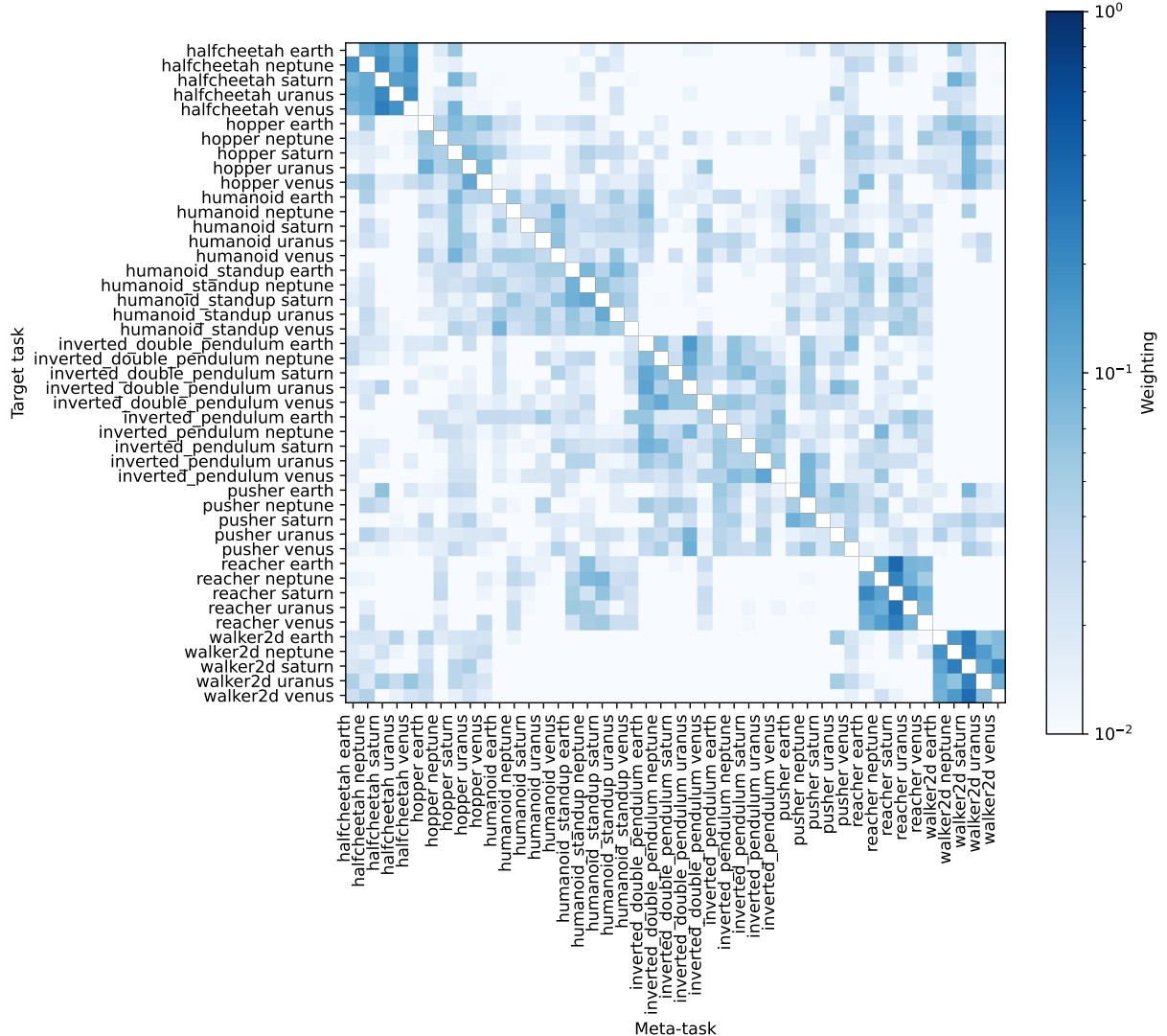

Figure 3: Average RGPE weights of MultiTaskPB2 for each task and all meta-tasks on BRAX. The weights form a diagonal block structure. Environments constructed from the same base environments with different gravity are identified as similar. We also see higher weights between different base environments that are related, for example, *humanoid* and *humanoid_standup*.

performance improvement. On the other hand, warmstarting had no substantial effect on MultiTaskPB2's final performance while slightly improving anytime performance in a statistically insignificant way. Overall, outside of TAFPB2, we did not observe a strong positive impact on the final performance. Given that there is no notable difference between the warmstarted TAFPB2 and both MultiTaskPB2 versions, we recommend MultiTaskPB2 to avoid additional complexity.

## 5.5 RQ4: Portfolio Training Interval

In this section, we investigate the empirical validity of our theoretical recommendation for the number of perturbation intervals we train configurations for during portfolio construction, as outlined in Section 4.4.

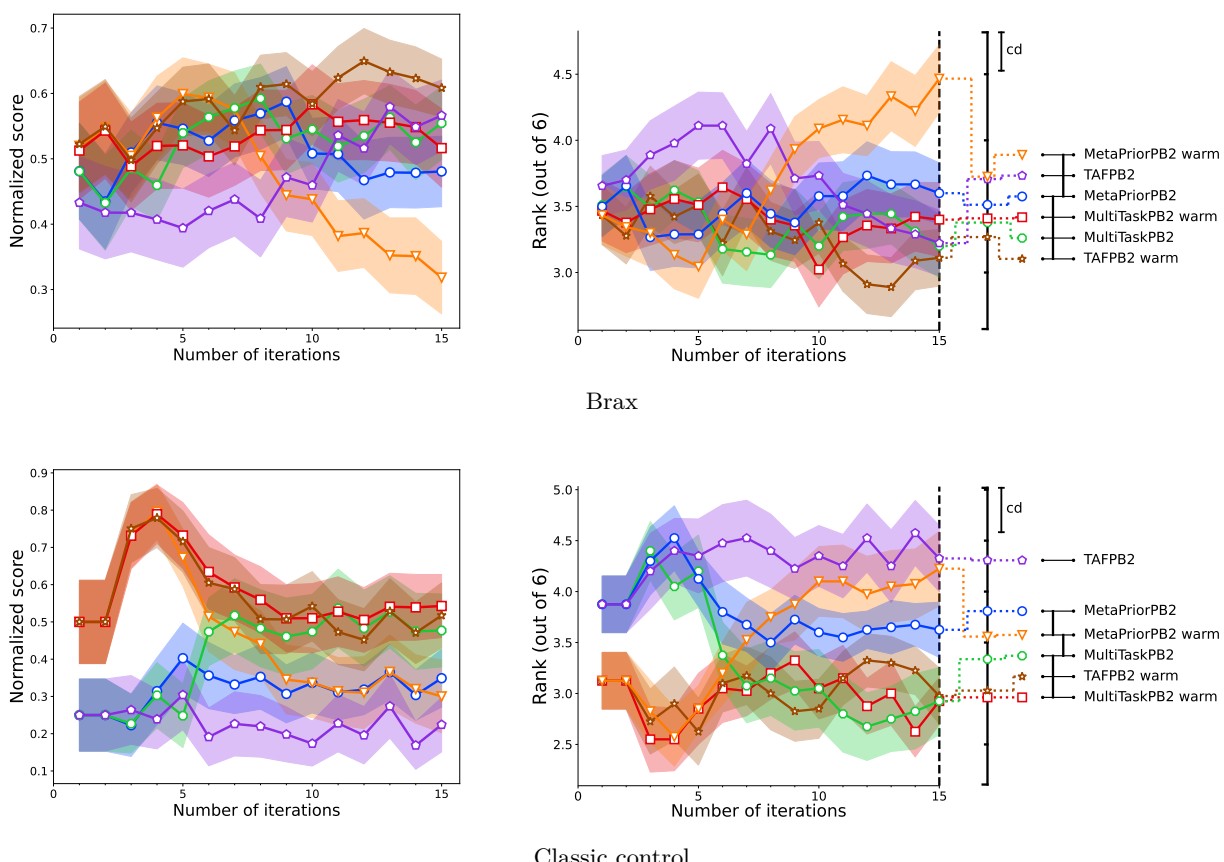

Figure 4: Warmstarting the meta-methods for PPO training on Brax (top) and classic control (bottom). Comparing rewards normalized at each iteration (higher is better) on the left and ranks on the right (lower is better). Warmstarted methods have better initial performance, but this effect is smaller in the more diverse Brax environments. Warmstarted and normal MultitaskPB2 perform the best overall.

We perform an ablation study comparing the performance of PB2* warmstarted with portfolios that were constructed using different numbers of perturbation intervals. We maintain the Ray default value for the fraction of copied workers, $\lambda = 0.25$. Following our theoretical argumentation from Section 4.4, we train for 4 perturbation intervals. We compare this training time with the original full training for 15 intervals and 1, 6, and 8 intervals.

Considering the computational demands of the portfolio generation process, we conducted this experiment on classic control tasks. The warmstarting was repeated over 10 seeds, and the configuration performance was averaged over 10 seeds during portfolio construction to ensure robustness.

As shown in Figure 5, PB2* warmstarted with the portfolio following our recommendations outperforms all other variants. From the third perturbation interval onwards, it achieves the best normalized score and rank, with statistically significant better anytime performance compared to all alternatives.

## 5.6 RQ5: Runtime Comparison

While our methods improve upon PB2, they also introduce an additional computational overhead. In this section, we take a closer look at the runtime of the methods we compare in Section 5.2. We ensured efficient training of the methods by matching the number of used CPU cores to the number of workers. For classic control experiments, we used machines with Intel Xeon Gold 6242 processors at 2.80 GHz, whereas for Brax,

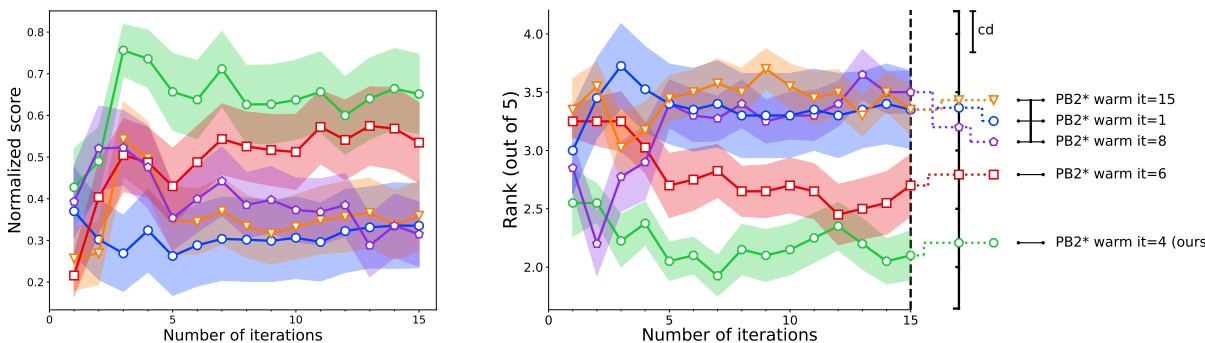

Figure 5: Comparing the performance of warmstarting PB2* for PPO training with different portfolios where configuration performance was calculated at a different number of perturbation intervals. Rewards that are normalized at each iteration (higher is better) are on the left and ranks are on the right (lower is better), results are on classic control. Warmstarting PB2* with the portfolio based on our theoretically motivated recommendation performs best.

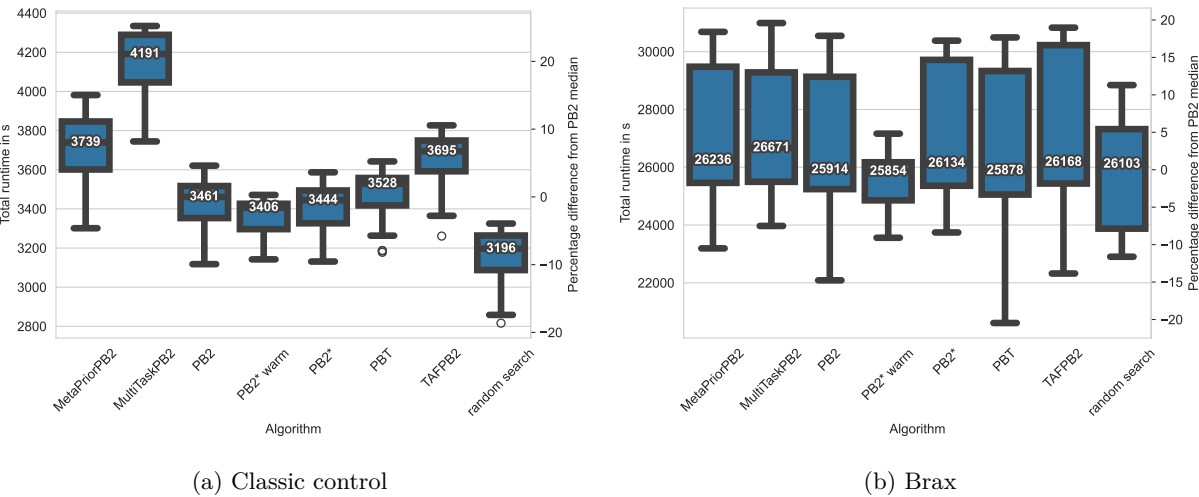

(a) Classic control

(b) Brax

Figure 6: Runtime for training PPO with different HPO methods on classic control and Brax. The overhead introduced by our methods becomes negligible for more complex environments with longer training times such as Brax.

we utilized machines with Intel Xeon E5-2630v4 processors at 2.2 GHz. For every method, we calculate the average runtime on each environment and visualize them as a boxplot in Figure 6. We can see that MultiTaskPB2, our best-performing method, introduces the biggest overhead and has the longest runtime. On classic control, the median runtime difference between MultiTaskPB2 and PB2 was 730 seconds, which amounts to a relative difference of 21%. On Brax the mean runtime difference is similar with 757 seconds, but the relative difference only amounts to 2.9%. The overhead of our methods when learning from up to 44 meta tasks where each task consists of the schedules of a combined 40 workers is roughly 12 minutes and becomes negligible when training in harder environments that might take multiple days to train. However, it is important to note that smaller organizations with limited computational resources may not benefit from our methods. Such organizations often cannot afford to run experiments on larger environments where the overhead becomes negligible. For them, the additional computational cost may outweigh the performance improvements, making our methods less practical for their specific needs. Thus, while our approach offers advantages in large-scale settings, it may not be suitable for smaller-scale experiments.

## 6   Conclusion

In this paper, we introduce new meta-learning approaches tailored for dynamic hyperparameter optimization in reinforcement learning (RL), aimed at enhancing the Population-Based Bandits (PB2) algorithm. PB2, leveraging time-varying Gaussian processes (GP), demonstrates strong overall performance but faces challenges when starting the optimization due to its GPs lacking information at the outset. To address this, we introduced four novel meta-learning methods for PB2 in time-varying environments. Our approaches include TAFPB2, which applies meta-learning to the acquisition function; MultiTaskPB2, employing multi-task GPs to enhance the surrogate model; MetaPriorPB2, utilizing a GP prior based on historical ensemble optimizations; and static hyperparameter portfolios for accelerated initialization. Among these, MultiTaskPB2 emerged as particularly promising, surpassing all other methods in both anytime and final performance metrics across two distinct RL environment families. Our contributions are notable for adapting meta-learning strategies to suit the dynamic nature of RL hyperparameter optimization.

While our findings show promise, our methods are constrained by two limitations. They depend on the availability of pertinent meta-data aligned with the hyperparameter search spaces and currently support only continuous hyperparameters. Moreover, implementing our methods entails a consistent computational overhead, which may hinder their suitability for environments with limited training durations. This disproportionately affects smaller organizations with limited computational resources that can not train on complex environments. Looking forward, future research could extend our methods to accommodate categorical hyperparameters, leveraging approaches similar to those explored by Parker-Holder et al. (2021) for PB2. Exploring more data-intensive meta-learning techniques, such as deep kernels (Arango et al., 2024) or prior-data fitted networks (PFNs) (Lee et al., 2024), might further enhance performance. Additionally, integrating meta-features for configuration selection during warmstarting and adapting our methods to domains like Deep Learning, where hyperparameters tend to be static, represent promising avenues for exploration.

### Acknowledgments

We acknowledge funding by the Deutsche Forschungsgemeinschaft (DFG, German Research Foundation) under SFB 1597 (SmallData), grant number 499552394. The authors acknowledge funding through the research network "Responsive and Scalable Learning for Robots Assisting Humans" (ReScaLe) of the University of Freiburg. The ReScaLe project is funded by the Carl Zeiss Foundation. The authors acknowledge support by the state of Baden-Württemberg through bwHPC and the German Research Foundation (DFG) through grant no INST 39/963-1 FUGG (bwForCluster NEMO). The authors acknowledge the financial support of the Hector Foundation.

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

# A Additional Technical Details

## A.1 Bayesian Optimization with GP

Bayesian optimization (BO) (Garnett, 2023; Snoek et al., 2012) is a popular sample-efficient black-box optimization technique that has been shown to outperform manual hyperparameter optimization . BO sequentially optimizes hyperparameters by constructing a probabilistic surrogate model from these observations to model the expensive target function. The probabilistic model is used to define an acquisition function that balances between exploration and exploitation for selecting the next hyperparameter configuration.

Gaussian Processes (GP) Rasmussen & Williams (2006) are a popular choice for the surrogate function. A GP is a stochastic process consisting of a set of random variables where every finite subset is jointly Gaussian. It is fully defined by its mean function $m \colon \mathcal{X} \to \mathbb{R}$ and positive definite covariance or kernel function $k \colon \mathcal{X} \times \mathcal{X} \to \mathbb{R}$. If a function $f$ follows a GP, we write

$$f \sim \mathcal{GP}(m, k).$$

Given a input $\mathbf{X} \coloneqq \{x_i \in \mathcal{X}\}_{i=1}^N$, with labels $\mathbf{y} \coloneqq \{y_i \in \mathbb{R}\}_{i=1}^N$ and a new point $x \in \mathcal{X}$, the posterior distribution $f_*$ at point $x$ is Gaussian

$$f_* \mid x, \mathbf{X}, \mathbf{y} \sim \mathcal{N}\left(\mu(x), \sigma^2(x)\right),$$

with the predictive mean $\mu(x)$ and covariance $\sigma^2(x)$, defined in closed form, assuming zero mean for simplicity

$$
\begin{aligned}
\mu(x) &= \mathbf{k}(x)^T (\mathbf{K} + \eta \mathbf{I})^{-1} \mathbf{y} \\
\sigma^2(x) &= k(x, x) - \mathbf{k}(x)^T (\mathbf{K} + \eta \mathbf{I})^{-1} \mathbf{k}(x),
\end{aligned}
\tag{2}
$$

where $\mathbf{k}(x) = [k(x, x_i)]_{i=1}^N$, $\mathbf{K} = [k(x_i, x_j)]_{i,j=1}^N$, and $\eta \geq 0$ is a noise term accounting for uncertainty in the data. For a more complete introduction to Gaussian Processes, we recommend the classic textbook by Rasmussen & Williams (2006).

## A.2 Population Based Bandits (PB2) Algorithm

---
**Algorithm 2** PB2
---
**Input:** Maximum number of perturbation intervals $T \in \mathbb{N}$, number of workers $W$, fractions of workers that *exploit* and *explore* per iteration $\lambda \in [0, 0.5]$
 1: $\mathcal{D} \leftarrow \emptyset, \mathcal{R} \leftarrow \emptyset$
 2: Initialize all workers $w = 1, 2, \ldots, W$ with model weights and hyperparameters $x_w$ and start training for $T$ perturbation intervals.
 3: **while** there is a worker training **do**
 4:    **if** worker $w$ reaches interval $t$ **then**
 5:       $\mathcal{D} \leftarrow \mathcal{D} \cup \left\{ \left( (t, \gamma_w, x_w), y_w \right) \right\}$                 ▷ Update observed data
 6:       **if** worker performance in bottom $\lambda$ fraction **then**
 7:          Copy model from random top $\lambda$ fraction of workers at timestep $t$ with initial performance $\bar{\gamma}$
 8:          Search for new configuration following Eq. (2)
 9:       **end if**
10:       Update set of running configurations $\mathcal{R}$
11:    **end if**
12: **end while**
---

## A.3 The expected improvement acquisition function with time-varying component

We consider a time-varying version of Expected Improvement (Močkus, 1974, EI) which takes into account the time-varying index $t$ as the additional input dimension. This time-varying EI is used with the TAFPB2

method described in Section 4.1.

$$\text{EI}_{\bar{t},\bar{\gamma}}(x, \mathcal{D}) = \left(\mu(\bar{t}, \bar{\gamma}, x) - f^*\right)\Phi\left(\frac{\mu(\bar{t}, \bar{\gamma}, x) - f^*}{\sigma(\bar{t}, \bar{\gamma}, x)}\right) + \sigma(\bar{t}, \bar{\gamma}, x)\varphi\left(\frac{\mu(\bar{t}, \bar{\gamma}, x) - f^*}{\sigma(\bar{t}, \bar{\gamma}, x)}\right),$$

where $f^* = \max_{((t,\gamma,x),y)\in\mathcal{D}} y$ is the best observed performance improvement at a time step $\bar{t}$.

### A.4 Ranking-Weighted GP Ensemble

In this section, we present the full RGPE weighting procedure from Section 4.1. To calculate the weighting, no meta-features describing the tasks are needed. The weights are based on how well a surrogate model $f(z) \sim \mathcal{N}(\mu(z), \sigma^2(z))$ where $z = \{t, \gamma, x\}$, describes the observed configurations $\mathcal{D}^{\tau+1}$ on the target task. This is measured by a pairwise ranking loss between predicted and real performances

$$\mathcal{L}(f, \mathcal{D}_{\tau+1}) = \sum_{\substack{(z_i, y_i), \\ (z_j, y_j)\in\mathcal{D}_{\tau+1}}} \mathbb{1}\Big(\left(\mu(z_i) < \mu(z_j)\right) \oplus (y_i < y_j)\Big), \tag{3}$$

where $\mathbb{1} : \{true, false\} \to \{0, 1\}, \{true \mapsto 1, false \mapsto 0\}$ and $\oplus$ is the exclusive OR operator (XOR). This loss estimates how well the target model fits the target data leads to overfitting since it would predict values for configurations it used for training. To prevent this, the Feurer et al. (2022b) use leave-one-out models $f_{\tau+1}^{-i}(z) \sim \mathcal{N}(\mu_{\tau+1}^{-i}(z), \sigma_{\tau+1}^{-i}{}^2(z))$ that fit $\mathcal{D}_{\tau+1} \setminus \{z_i, y_i\}$. Using these models, the pairwise ranking loss for the target task is calculated as:

$$\mathcal{L}(f_{\tau+1}, \mathcal{D}_{\tau+1}) = \sum_{\substack{(z_i, y_i), \\ (z_j, y_j)\in\mathcal{D}_{\tau+1}}} \mathbb{1}\Big(\left(\mu_{\tau+1}^{-i}(z_i) < y_j\right) \oplus (y_i < y_j)\Big). \tag{4}$$

Using a ranking loss over, for example, the mean squared error is preferable, since it better matches the goal of optimizing the hyperparameters. For optimization, only the ordering of the hyperparameters is relevant. These ranking losses are used to calculate the weights as seen in Algorithm 3.

---

**Algorithm 3** RGPE weighting

---

**Input:** Number of bootstrap samples $S$, observations on the target task $\mathcal{D}_{\tau+1}$, meta-models $f_i$ for $i = 1, 2, \ldots, \tau$ and the target model $f_{\tau+1}$
**Output:** Model weighting $w_i$ for $i = 1, 2, \ldots, \tau + 1$

1: **for** $s = 1, \ldots, S$ **do**                                                         ▷ Calculate losses
2:     $\mathcal{D}_{\tau+1}^s \leftarrow \texttt{bootstrap}(\mathcal{D}_{\tau+1})$
3:     **for** $i = 1, \ldots, \tau$ **do**
4:         $l_{i,s} \leftarrow \mathcal{L}\left(f_i, \mathcal{D}_{\tau+1}^s\right)$                                         ▷ Using Eq (3)
5:     **end for**
6:     $l_{\tau+1,s} \leftarrow \mathcal{L}\left(f_{\tau+1}, \mathcal{D}_{\tau+1}^s\right)$                                ▷ Using Eq. (4)
7: **end for**
8: **for** $i = 1, \ldots, \tau + 1$ **do**                                           ▷ Calculate weights
9:     $w_i \leftarrow \frac{1}{S}\sum_{s=1}^{S}\left(\frac{\mathbb{1}\left(i\in\arg\min_{i'} l_{i',s}\right)}{\sum_{j=1}^{t}\mathbb{1}\left(j\in\arg\min_{i'} l_{i',s}\right)}\right)$
10: **end for**

---

## B Detailed Results

### B.1 Detailed Experimental Setup

**Search Space** For the experiments, we use the same PPO search space as Parker-Holder et al. (2020a), the only difference being that we fix the batch size to better control the memory usage. The batch size is fixed to 20.000 and 25.000 for classic control and Brax, respectively. The search space is shown in Table 1.

Table 1: **Hyperparameter ranges PPO.**

| Hyperparameter | Range |
| --- | --- |
| Learning rate | (1e-5, 1e-3) |
| Lambda | (0.9, 0.99) |
| Clip parameter | (0.1, 0.5) |

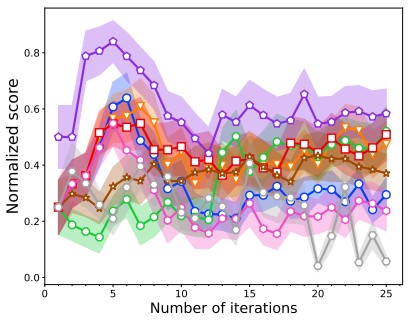 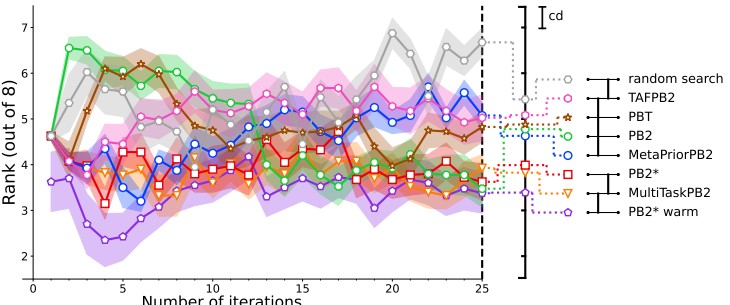

Figure 7: Training all methods on classic control for 1,000,000 steps instead of 600,000 with a perturbation interval of 60,000 steps. Comparing rewardsnormalized at each iteration (higher is better) on the left and ranks on the right (lower is better). Using meta-learning later in training when there is enough data to describe the target task is less beneficial. Still, MultiTaskPB2 and warmstarted PB2* remain the top-performing methods for training PPO on classic control.

**Hyper-hyperparameter settings** The methods we propose have multiple hyper-hyperparameters whose values we set for the experiments. For the RGPE weight computation, we sampled 100 bootstraps, and for MultiTaskPB2, we used the weights to restrict ourselves to the 4 most similar tasks. From each of these meta-task we use 100 datapoints to train the multi-task GP. During the portfolio construction, we generated the portfolio candidates via random search. On classic control, we selected the best 2 out of 30 random configurations, and on Brax, we selected the best out of 20. The performance is averaged over 10 seeds.

**Environments** We used CARL (Benjamins et al., 2023) to vary the gravity of the 4 classic control environments (*mountain_car, cart_pole, pendulum, acrobot*) and the 9 Brax environments (*humanoid, halfcheetah, hopper, humanoid_standup, inverted_double_pendulum, inverted_pendulum, pusher, reacher, and walker2d*). Since CARL does not allow us to vary the gravity in all of the classic control environments, we instead altered the mass of the robots in such environments by multiplying with the relative gravity of the planets with respect to Earth. The different gravity values can be seen in Table 2. For classic control, we trained the agents on each environment for 600,000 steps with a perturbation interval of 40,000. The agents on the Brax environments were each trained for 3,000,000 steps with a perturbation interval of 200,000. We also looked at increasing the number of training steps and trained classic control for 1,000,000 steps in Figure 7. Such an ablation was too expensive for Brax.

Table 2: **Gravity variation for the RL environments.** Values retrieved from `https://nssdc.gsfc.nasa.gov/planetary/factsheet/index.html`

| Planet | Uranus | Venus | Saturn | Earth | Neptune |
| --- | --- | --- | --- | --- | --- |
| Gravity | 8.7 | 8.9 | 9.0 | 9.81 | 11.0 |

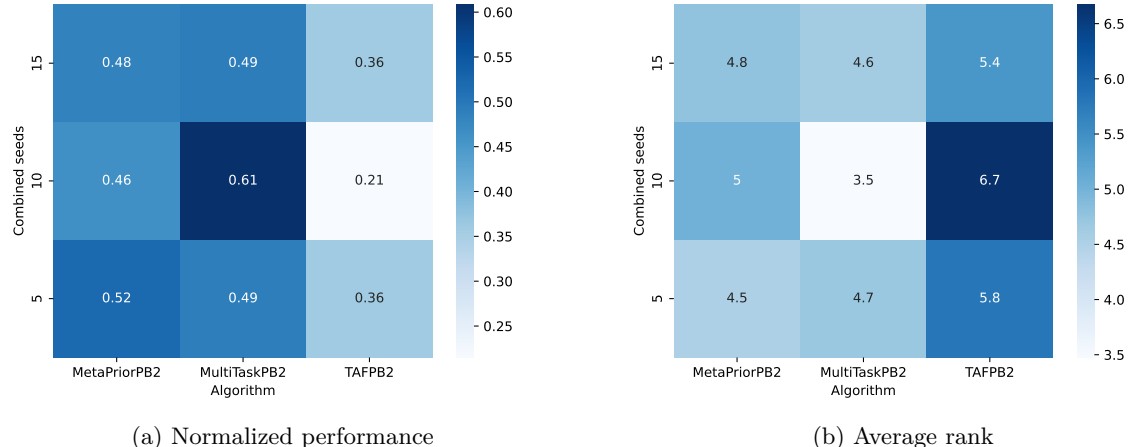

(a) Normalized performance              (b) Average rank

Figure 8: Performance at the end of training PPO on classic control for our meta-methods based on how many seeds were used to make up the meta-data. On the left is normalized performance (higher is better) and on the right is rank (lower is better).

## B.2 Meta-data Requirements

In our experiments, we combined runs of PB2, using ten different seeds to create the meta-data. In this section, we aim to determine whether this amount of meta-data is sufficient and if there is any benefit to using more seeds. To explore this, we examine the performance of TAFPB2, MetaPriorPB2, and MultiTaskPB2 when the meta-data consists of combinations of five, ten, and fifteen runs.

The results, presented in Table 8, reveal no significant trend in the data. Of all the methods, MultiTaskPB2, using meta-data that combined the runs of 10 seeds, performed the best. However, it is possible that MultiTaskPB2 might benefit from more meta-data if we adjust its hyper-hyperparameters, especially the number of points that are input to the multi-task GP from each meta-task.

## B.3 General vs Individual Portfolios

When analyzing the warmstarted and vanilla versions of PB2* in Section 5.2, we observe that warmstarting does not significantly affect performance on Brax, but it did enhance initial performance on classic control. This difference may be due to the broader range of environments that Brax encompasses, making it harder for a general portfolio to optimize effectively. Using meta-features to select an individual portfolio for each task is a way to avoid this limitation. In this section, we compare the performance of individual portfolios with our general portfolio. We employ the same algorithm outlined in Section 4.4 to construct individual portfolios, but we limit the performance matrix to environments sharing the same base environment, thereby avoiding the use of meta-features.

Figure 9 shows the visualizations of the two portfolio types on Brax and classic control. Note that in our general portfolio approach, we also have different portfolios for each environment; these portfolios are not optimized for the given environment but are a result of our leave-one-out strategy that excludes the performance of all configurations on the target environment. In both environment families, we see that the general portfolios often do not contain the best configurations. Examples include *pendulum* on classic control and *walker2d*, *hopper*, and *inverted_double_pendulum* on Brax. In contrast, the individual portfolios encompass most of the top-performing configurations.

The impact of these differences in portfolios on HPO performance is illustrated in Figure 10. We compare PB2* warmstarted with our individual portfolio to PB2* warmstarted with our general portfolio, PB2*, and our overall best approach MultiTaskPB2. On classic control, using the individual portfolio results in an

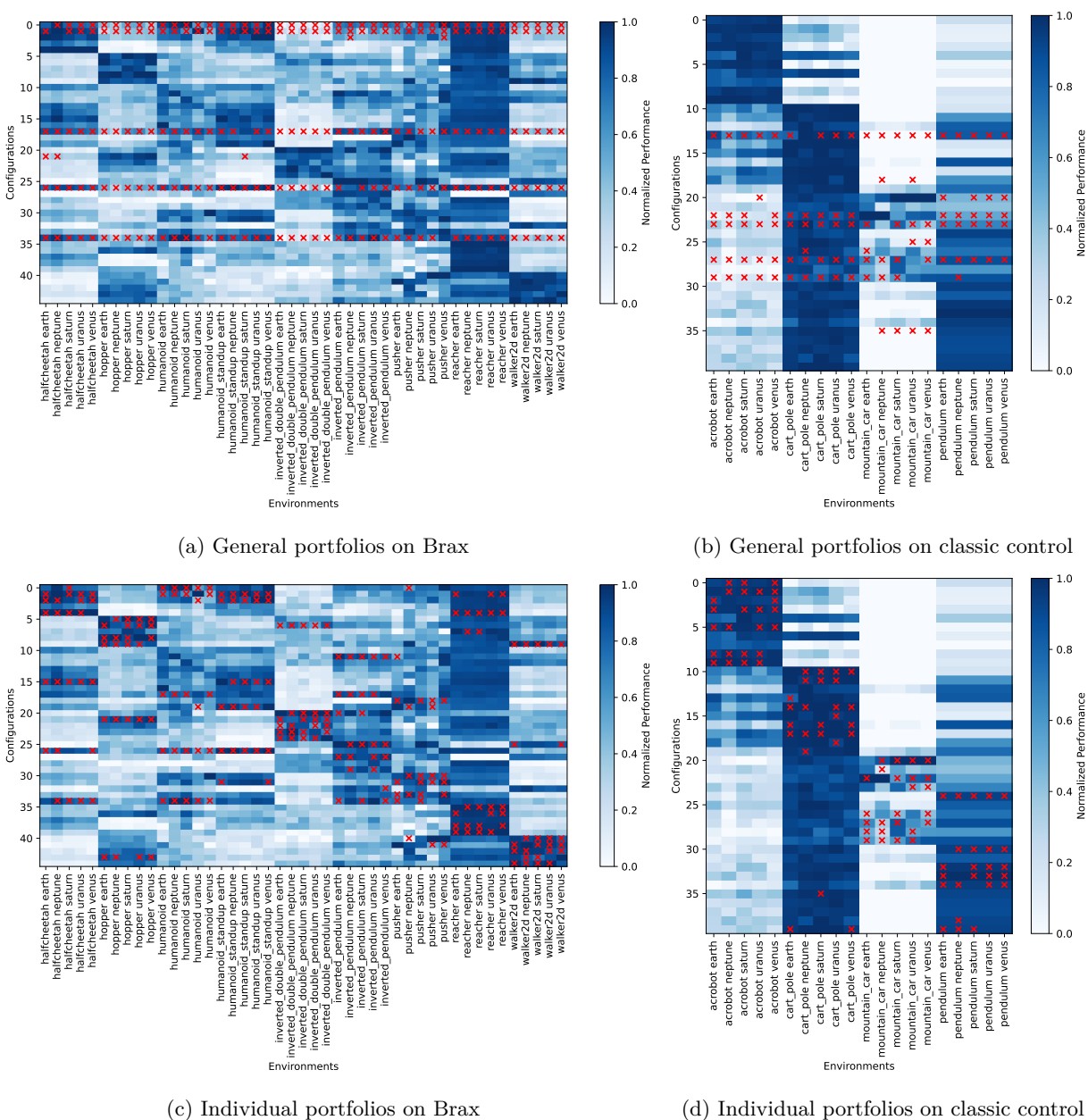

(a) General portfolios on Brax

(b) General portfolios on classic control

(c) Individual portfolios on Brax

(d) Individual portfolios on classic control

Figure 9: Portfolio performance matrix for Brax (left) and classic control (right). Red crosses signal that the marked configuration is part of the portfolio for the environment. We observe that the portfolios do not contain the best configurations for multiple environments (e.g. *pendulum* on classic control and *walker2d*, *inverted_double_pendulum* on Brax). This is remedied by constructing portfolios based on the performance of the configurations on the same base environment (bottom).

even greater initial performance boost compared to the general portfolio, this does not translate into better final performance. On Brax, warmstarting with the individual portfolio leads to a significant improvement in initial performance that continues through the optimization process, achieving the best final performance among all approaches. These findings suggest that future studies should investigate the potential of using meta-features to warmstart population-based optimization methods.

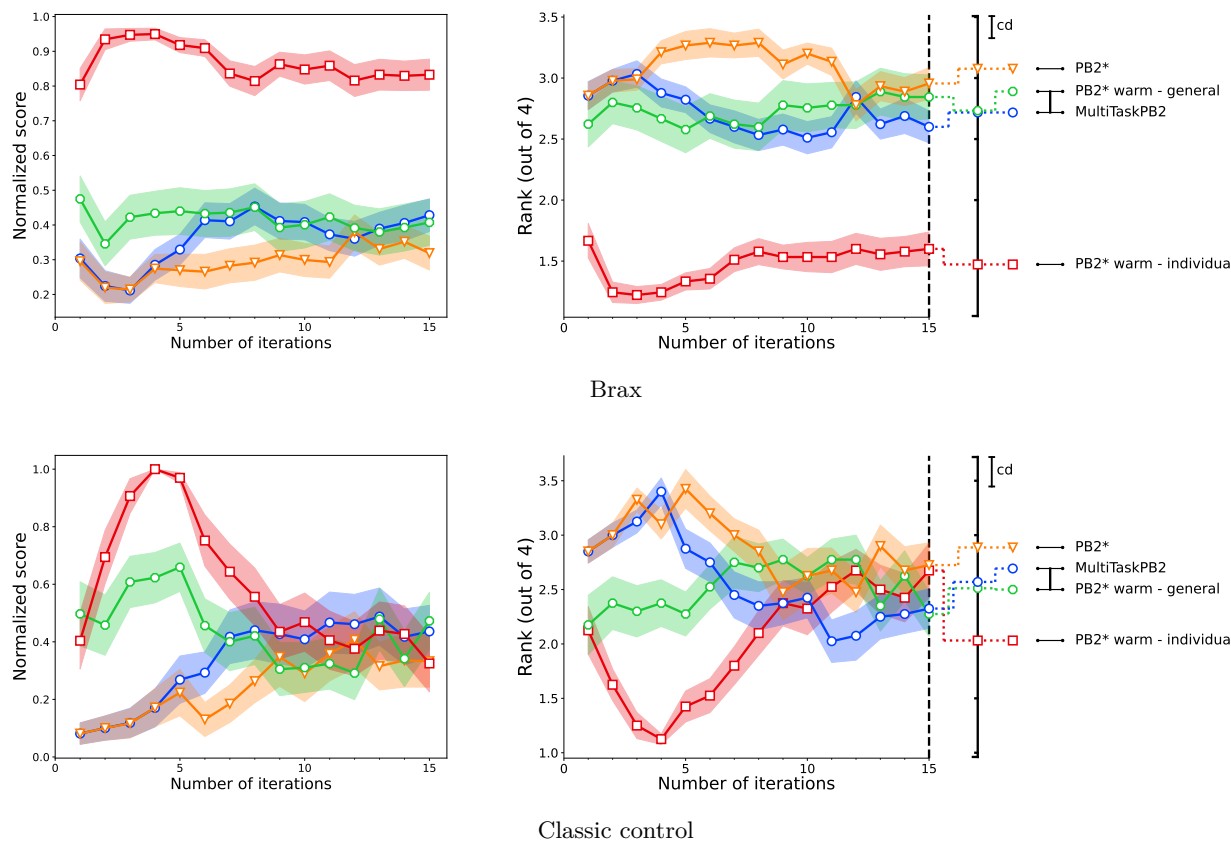

Figure 10: Performance comparison between using individual and general portfolios for warmstarting on Brax (top) and classic control (bottom). Warmstarting with an individual portfolio is statistically significantly better than all other methods for anytime performance. On Brax individual portfolios are also the best final performers. On the other hand, on classic control, individual portfolios have worse final performance compared to a general portfolio.

## B.4  Additional Similarity Weighting Results

In Section 5.3, we analyze the behavior of MultiTaskPB2 by examining the distribution of RGPE weights on Brax. Here, we provide complementary visualizations for classic control environments and for the MetaPriorPB2 and TAFPB2 methods. The visualizations for MetaPriorPB2 on Brax are shown in Figure 11, and those for TAFPB2 are shown in Figure 12. The results for all three methods on classic control environments are presented in Figure 13.

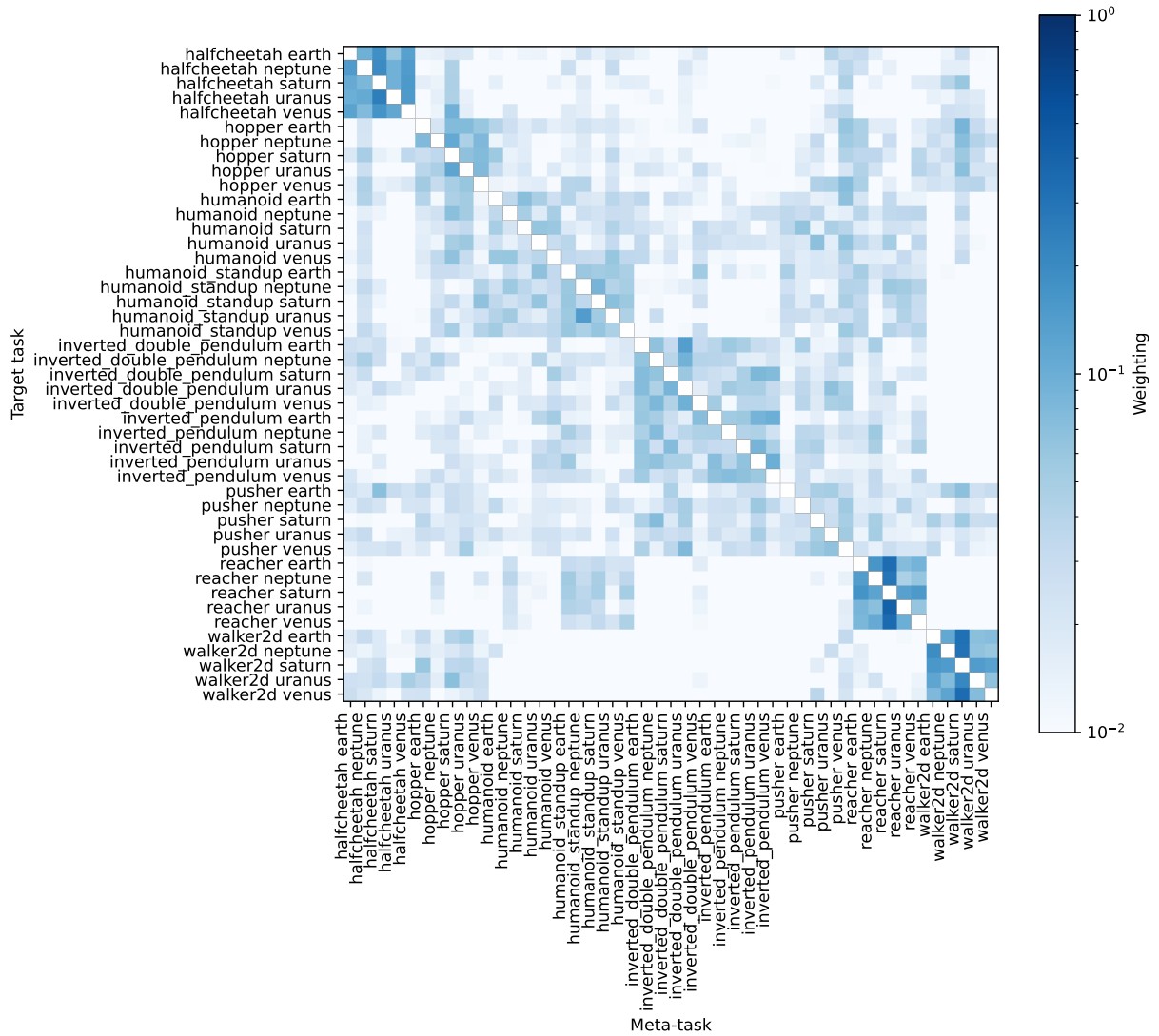

Figure 11: Average RGPE weights of MetaPriorPB2 for each task and all meta-tasks on Brax. The results are in line with MetaPriorPB2.

## B.5   Ablation over the Number of Workers

In our other experiments, we consistently used four parallel workers. However, in this section, we expand our analysis to consider the use of eight and twelve workers. We compare our methods and the baselines for these different numbers of workers in Figure 14.

Our results show that MultiTaskPB2 performs well overall, maintaining strong performance across different worker counts. MetaPriorPB2 shows improvement and becomes comparable to MultiTaskPB2 as the number of workers increases. Notably, when utilizing twelve workers, the warmstarting approach demonstrates the strongest performance.

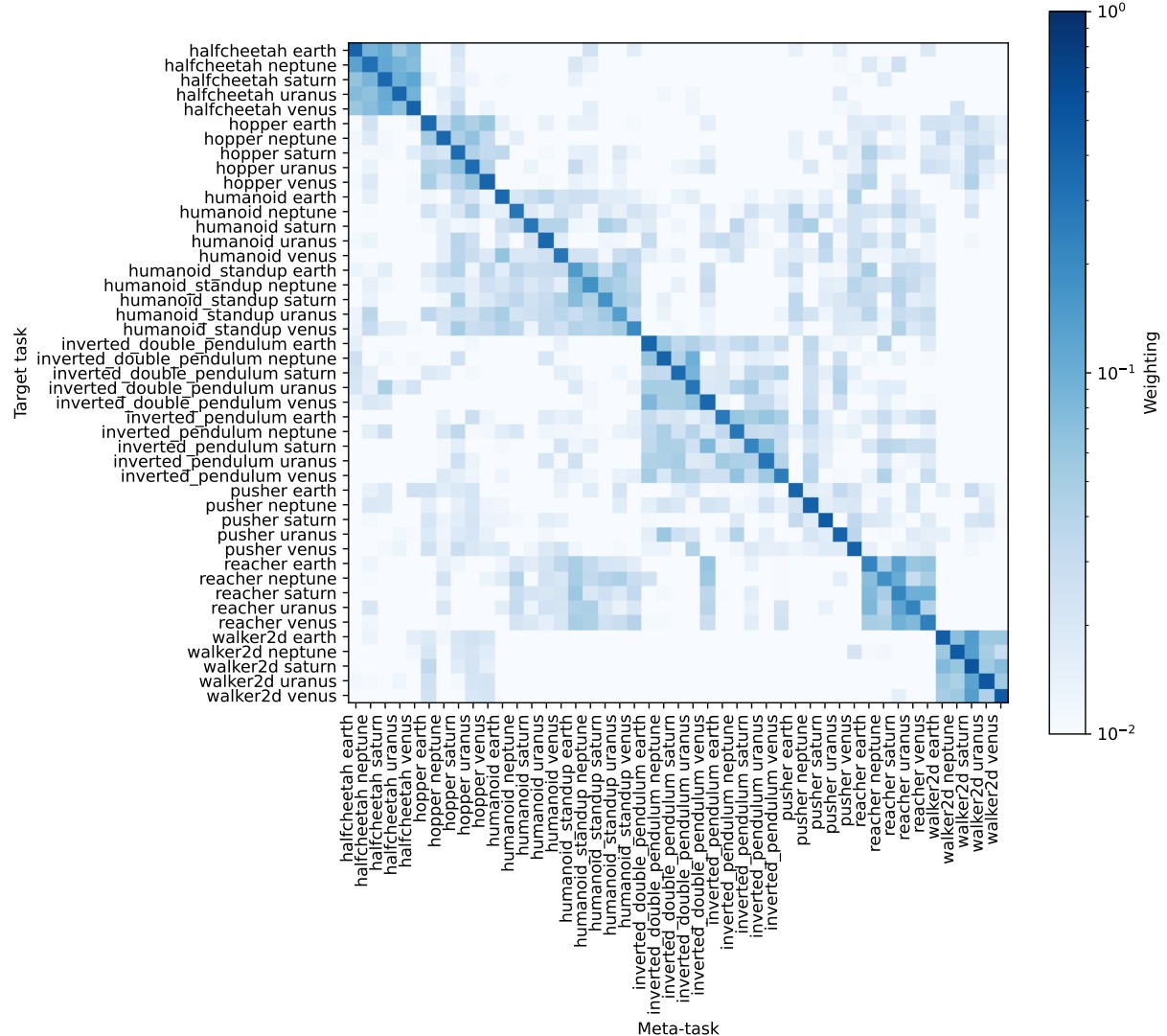

Figure 12: Average RGPE weights of TAFPB2 for each task and all meta-tasks on Brax. The results are in line with MetaPriorPB2. Note that the diagonal is not filled with nan's since TAFPB2 computes how well a GP trained on data from the target task fits the target task.

## B.6 Crashed Runs on Brax

During our experiments, some training runs crashed on Brax due to divergence issues. The primary cause of these crashes is the choice of our hyperparameter ranges, which were designed for classic control by Parker-Holder et al. (2020a) and are less suitable for Brax.

Table 3 presents a summary of the crashed runs. PB2 was the most affected, with 27 total crashes, while random search, MultiTaskPB2, and MetaPriorPB2 experienced the fewest crashes, with 3, 3, and 4 crashes, respectively. The lower crash rate for random search is due to its use of constant hyperparameters. Multi-

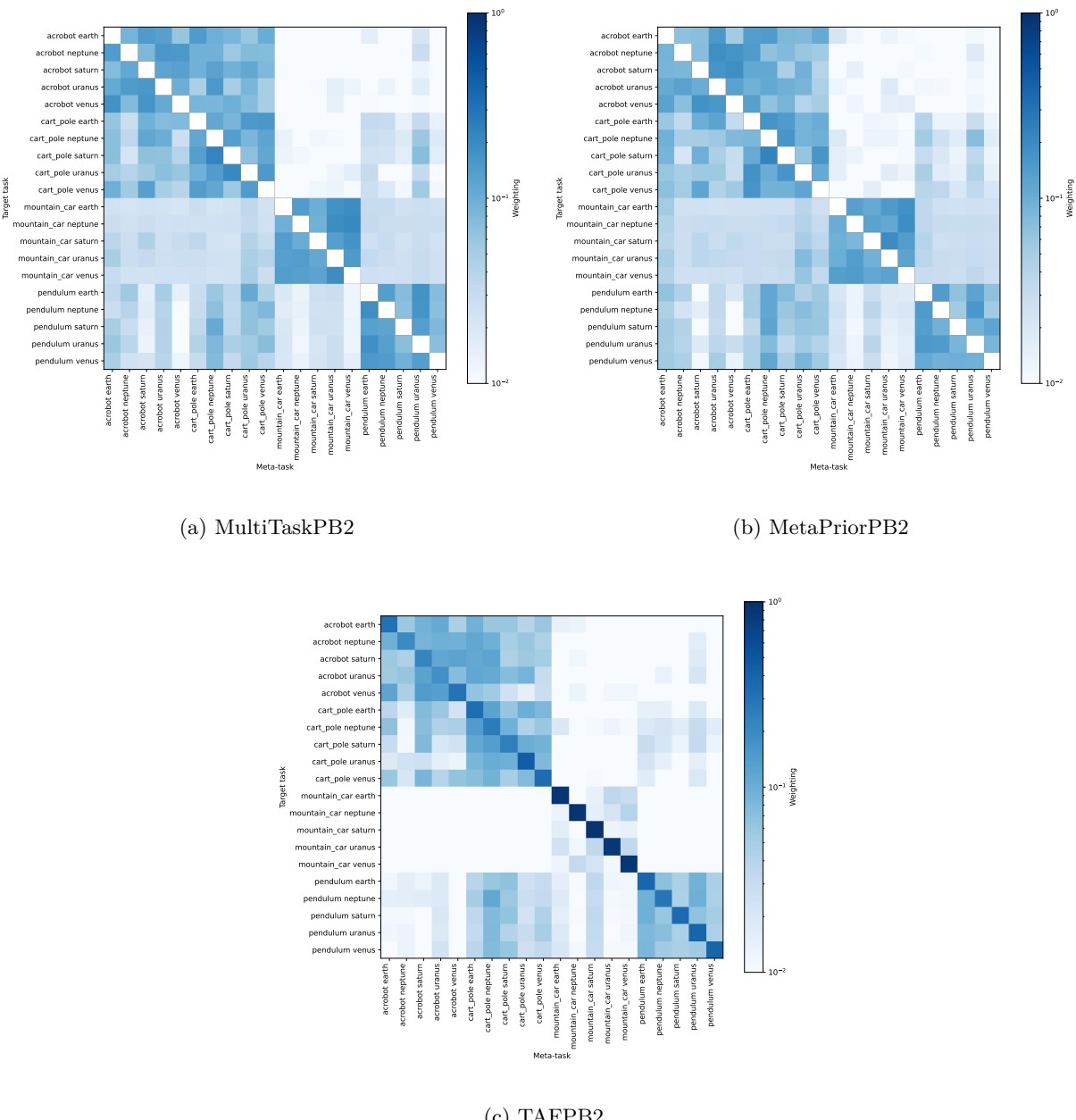

Figure 13: Average RGPE weights of TAFPB2, MetaPriorPB2, and MultiTaskPB2 for each task and all meta-tasks on classic control. Like for Brax, the weights form a diagonal block structure. Bigger blocks can be seen between *acrobot* and *cart_pole* and to a degree between *cart_pole* and *pendulum*.

TaskPB2 and MetaPriorPB2's ability to avoid detrimental hyperparameter regions showcases their robustness, supporting the reliability of our approaches.

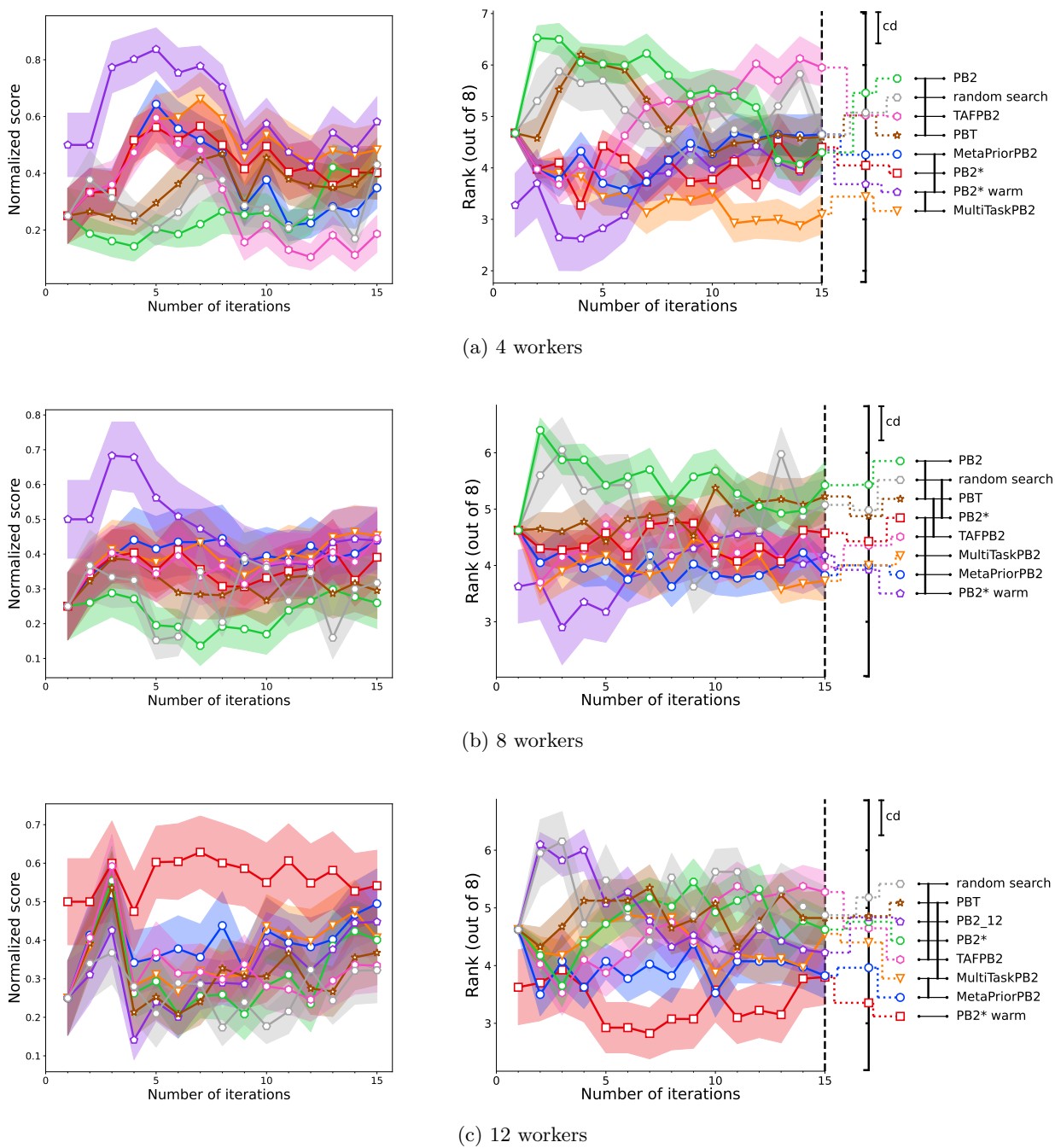

Figure 14: Ablation over the number of workers on classic control. The top shows results for 4 workers, the middle for 8 and the bottom for 12 workers. MultiTaskPB2 shows strong performance for all numbers of workers while warmstarting PB2* and MetaPriorPB2 have a relative benefit in performance with increased number of workers.

Table 3: Crashed runs algorithms

| Algorithm Environment | MetaPriorPB2 | MultiTaskPB2 | PB2 | PB2* | PB2* warm | PBT | TAFPB2 | random search |
|---|---|---|---|---|---|---|---|---|
| halfcheetah earth | 0 | 0 | 0 | 0 | 0 | 0 | 0 | 0 |
| halfcheetah neptune | 0 | 0 | 0 | 0 | 0 | 0 | 0 | 0 |
| halfcheetah saturn | 0 | 0 | 0 | 0 | 0 | 0 | 0 | 0 |
| halfcheetah uranus | 0 | 0 | 0 | 0 | 0 | 0 | 0 | 0 |
| halfcheetah venus | 0 | 0 | 0 | 0 | 0 | 0 | 0 | 0 |
| hopper earth | 0 | 0 | 0 | 0 | 0 | 0 | 0 | 0 |
| hopper neptune | 0 | 0 | 0 | 0 | 0 | 0 | 0 | 0 |
| hopper saturn | 0 | 0 | 0 | 0 | 0 | 0 | 0 | 0 |
| hopper uranus | 0 | 0 | 0 | 0 | 0 | 0 | 0 | 0 |
| hopper venus | 0 | 0 | 0 | 0 | 0 | 0 | 0 | 0 |
| humanoid earth | 0 | 0 | 0 | 0 | 1 | 0 | 0 | 0 |
| humanoid neptune | 0 | 0 | 0 | 0 | 0 | 0 | 0 | 0 |
| humanoid saturn | 0 | 0 | 0 | 0 | 0 | 0 | 0 | 0 |
| humanoid uranus | 0 | 0 | 0 | 0 | 1 | 0 | 0 | 0 |
| humanoid venus | 0 | 0 | 0 | 0 | 0 | 0 | 0 | 0 |
| humanoid_standup earth | 0 | 0 | 1 | 0 | 0 | 0 | 0 | 0 |
| humanoid_standup neptune | 0 | 0 | 0 | 0 | 0 | 0 | 0 | 0 |
| humanoid_standup saturn | 0 | 0 | 0 | 0 | 0 | 0 | 0 | 0 |
| humanoid_standup uranus | 0 | 0 | 0 | 0 | 0 | 0 | 0 | 0 |
| humanoid_standup venus | 0 | 0 | 1 | 0 | 0 | 0 | 0 | 0 |
| inverted_double_pendulum earth | 0 | 0 | 3 | 1 | 1 | 0 | 2 | 0 |
| inverted_double_pendulum neptune | 0 | 1 | 3 | 2 | 2 | 1 | 5 | 0 |
| inverted_double_pendulum saturn | 2 | 1 | 7 | 2 | 0 | 4 | 4 | 0 |
| inverted_double_pendulum uranus | 1 | 0 | 2 | 2 | 2 | 1 | 2 | 0 |
| inverted_double_pendulum venus | 1 | 0 | 4 | 4 | 1 | 1 | 3 | 0 |
| inverted_pendulum earth | 0 | 1 | 0 | 1 | 1 | 0 | 0 | 1 |
| inverted_pendulum neptune | 0 | 0 | 1 | 0 | 0 | 0 | 1 | 1 |
| inverted_pendulum saturn | 0 | 0 | 1 | 0 | 0 | 0 | 2 | 1 |
| inverted_pendulum uranus | 0 | 0 | 1 | 0 | 0 | 0 | 1 | 0 |
| inverted_pendulum venus | 0 | 0 | 2 | 0 | 0 | 2 | 0 | 0 |
| pusher earth | 0 | 0 | 0 | 0 | 1 | 0 | 0 | 0 |
| pusher neptune | 0 | 0 | 1 | 0 | 1 | 0 | 0 | 0 |
| pusher saturn | 0 | 0 | 0 | 0 | 1 | 0 | 0 | 0 |
| pusher uranus | 0 | 0 | 0 | 0 | 0 | 0 | 0 | 0 |
| pusher venus | 0 | 0 | 0 | 0 | 1 | 0 | 0 | 0 |
| reacher earth | 0 | 0 | 0 | 0 | 0 | 0 | 0 | 0 |
| reacher neptune | 0 | 0 | 0 | 0 | 0 | 0 | 0 | 0 |
| reacher saturn | 0 | 0 | 0 | 0 | 0 | 0 | 0 | 0 |
| reacher uranus | 0 | 0 | 0 | 0 | 0 | 0 | 0 | 0 |
| reacher venus | 0 | 0 | 0 | 0 | 1 | 0 | 0 | 0 |
| walker2d earth | 0 | 0 | 0 | 0 | 0 | 0 | 0 | 0 |
| walker2d neptune | 0 | 0 | 0 | 0 | 0 | 0 | 0 | 0 |
| walker2d saturn | 0 | 0 | 0 | 0 | 0 | 0 | 0 | 0 |
| walker2d uranus | 0 | 0 | 0 | 0 | 0 | 0 | 0 | 0 |
| walker2d venus | 0 | 0 | 0 | 0 | 0 | 0 | 0 | 0 |
| Sum | 4 | 3 | 27 | 12 | 14 | 9 | 20 | 3 |

### B.7 Raw Results

In this section, we display the raw rewards at the end of training for all environments and methods used in our main method comparison (Section 5.2). The rewards for classic control environments are detailed in Table 4, and those for Brax environments are listed in Table 5.

Table 4: Average reward and standard error at the end of training on classic control.

| algorithm dataset | MetaPriorPB2 | MultiTaskPB2 | PB2 | PB2* | PB2* warm | PBT | TAFPB2 | random search |
|---|---|---|---|---|---|---|---|---|
| acrobot earth | -79.40±1.28 | -79.90±2.27 | -80.18±1.61 | -78.18±1.04 | -79.88±1.27 | **-76.25±0.55** | -80.95±2.48 | -83.83±3.13 |
| acrobot neptune | -89.46±1.60 | -86.83±1.06 | -88.19±1.28 | -86.56±1.33 | -88.07±1.50 | **-85.21±1.09** | -87.17±1.17 | -88.29±2.05 |
| acrobot saturn | -73.39±1.78 | -74.85±1.72 | -74.01±1.43 | -74.84±2.56 | -74.21±1.64 | **-72.96±1.19** | -74.19±0.98 | -73.16±1.95 |
| acrobot uranus | -76.71±5.55 | -71.28±2.66 | -68.76±1.90 | -71.41±2.25 | **-68.75±1.50** | -69.90±1.72 | -70.33±1.45 | -71.76±1.56 |
| acrobot venus | -73.03±3.22 | -72.73±1.93 | -74.31±3.29 | -77.48±2.87 | -76.55±2.88 | -72.89±1.65 | -75.24±2.11 | **-71.91±1.65** |
| cart_pole earth | 500.00±0.00 | 500.00±0.00 | 500.00±0.00 | 500.00±0.00 | 500.00±0.00 | 500.00±0.00 | 500.00±0.00 | 500.00±0.00 |
| cart_pole neptune | 500.00±0.00 | 500.00±0.00 | 500.00±0.00 | 500.00±0.00 | 500.00±0.00 | 500.00±0.00 | 500.00±0.00 | 500.00±0.00 |
| cart_pole saturn | 500.00±0.00 | 500.00±0.00 | 499.48±0.52 | 500.00±0.00 | 500.00±0.00 | 500.00±0.00 | 500.00±0.00 | 500.00±0.00 |
| cart_pole uranus | 500.00±0.00 | 500.00±0.00 | 500.00±0.00 | 500.00±0.00 | 500.00±0.00 | 500.00±0.00 | 500.00±0.00 | 500.00±0.00 |
| cart_pole venus | 500.00±0.00 | 500.00±0.00 | 500.00±0.00 | 500.00±0.00 | 500.00±0.00 | 500.00±0.00 | 500.00±0.00 | 500.00±0.00 |
| mountain_car earth | -130.26±12.16 | -131.18±12.37 | -139.77±13.24 | -145.05±13.33 | **-100.48±1.54** | -137.77±11.70 | -146.69±12.32 | -145.50±9.05 |
| mountain_car neptune | -144.81±10.10 | -144.35±10.16 | -144.60±10.44 | -149.25±10.44 | **-114.13±2.11** | -149.98±10.50 | -146.49±9.64 | -152.21±9.73 |
| mountain_car saturn | -131.11±12.31 | -131.02±12.67 | -141.82±10.99 | -141.22±11.77 | **-98.10±1.44** | -141.14±10.64 | -140.23±12.01 | -138.59±13.39 |
| mountain_car uranus | -127.83±12.44 | -133.72±11.80 | -135.94±12.16 | -132.93±12.26 | **-98.51±2.97** | -137.08±12.46 | -134.23±12.14 | -129.13±13.12 |
| mountain_car venus | -134.05±12.37 | -133.00±12.47 | -134.63±12.87 | -135.30±12.82 | **-95.05±1.32** | -139.81±11.95 | -140.57±12.07 | -140.16±12.64 |
| pendulum earth | -242.98±18.59 | **-215.39±23.16** | -216.83±13.00 | -220.57±19.75 | -238.17±28.75 | -238.13±21.94 | -268.12±52.92 | -220.73±26.21 |
| pendulum neptune | -256.10±29.46 | -217.07±23.73 | -218.04±13.21 | -220.74±19.43 | **-211.48±28.79** | -252.17±29.85 | -295.98±48.61 | -220.73±26.21 |
| pendulum saturn | -227.04±25.29 | **-213.02±21.14** | -216.83±13.00 | -219.51±19.46 | -238.04±28.78 | -252.21±23.40 | -286.01±54.95 | -220.73±26.21 |
| pendulum uranus | -236.29±20.67 | **-217.74±20.71** | -224.42±14.93 | -218.48±19.54 | -248.17±34.88 | -252.45±24.39 | -271.76±45.85 | -220.73±26.21 |
| pendulum venus | -265.41±18.44 | **-212.04±21.48** | -216.83±13.00 | -218.48±19.54 | -238.08±28.79 | -226.41±9.43 | -280.74±49.15 | -220.73±26.21 |

Table 5: Average reward and standard error at the end of training on Brax.

| algorithm dataset | MetaPriorPB2 | MultiTaskPB2 | PB2 | PB2* | PB2* warm | PBT | TAFPB2 | random search |
|---|---|---|---|---|---|---|---|---|
| halfcheetah earth | 1298.24±63.37 | **1417.09±84.69** | 1369.25±77.61 | 1369.49±65.47 | 1266.56±47.40 | 1381.45±84.78 | 1286.77±90.79 | 1198.84±72.21 |
| halfcheetah neptune | **1447.59±77.68** | 1416.99±63.85 | 1440.36±59.74 | 1401.98±60.56 | 1272.20±35.37 | 1329.84±55.14 | 1406.62±58.67 | 1299.20±86.32 |
| halfcheetah saturn | **1351.66±106.11** | 1310.18±123.96 | 1293.64±99.07 | 1284.06±99.90 | 1108.08±26.35 | 1233.10±73.22 | 1252.88±139.29 | 1186.38±87.61 |
| halfcheetah uranus | 1211.14±109.14 | 1212.86±99.51 | 1229.33±132.43 | 1199.22±100.00 | 1181.52±49.39 | **1262.79±120.31** | 1198.47±111.37 | 1075.46±91.92 |
| halfcheetah venus | 1326.86±54.05 | 1312.72±46.91 | 1225.72±101.66 | 1294.17±66.66 | 1159.17±49.81 | 1241.12±92.76 | **1332.18±50.05** | 1110.89±85.70 |
| hopper earth | 660.27±23.86 | 726.93±19.07 | 685.93±34.00 | 682.43±27.23 | 690.67±28.61 | **752.84±23.22** | 710.67±22.77 | 646.35±24.43 |
| hopper neptune | 680.18±23.36 | 689.45±22.92 | 676.96±27.19 | 671.33±16.44 | 653.62±15.62 | **724.13±19.95** | 688.95±14.95 | 683.14±16.82 |
| hopper saturn | **745.99±32.34** | 719.04±25.27 | 723.02±19.00 | 718.91±22.67 | 702.23±29.30 | 706.41±40.01 | 697.23±24.30 | 728.69±20.88 |
| hopper uranus | 733.63±35.02 | 724.93±35.13 | 762.17±28.50 | **762.21±34.02** | 679.63±24.87 | 745.28±31.43 | 754.52±28.89 | 734.31±26.01 |
| hopper venus | 728.46±29.73 | 715.24±28.94 | 717.75±29.90 | 721.59±36.88 | 742.35±29.95 | 702.59±43.60 | **757.02±24.57** | 724.57±22.52 |
| humanoid earth | 404.77±9.42 | 379.64±6.48 | 385.83±5.88 | 386.08±8.56 | **415.66±6.02** | 393.57±10.86 | 394.75±9.60 | 383.76±4.01 |
| humanoid neptune | 368.57±7.88 | 361.79±10.42 | 369.60±8.40 | 371.90±10.50 | **402.83±5.76** | 369.13±6.17 | 356.29±9.35 | 370.82±5.94 |
| humanoid saturn | 403.81±9.54 | 416.16±10.57 | 403.66±10.70 | 409.13±13.29 | **436.46±8.80** | 420.11±6.56 | 397.12±7.07 | 398.29±4.78 |
| humanoid uranus | 421.16±13.51 | 402.81±10.92 | 406.51±9.29 | 415.79±8.94 | **431.10±13.96** | 412.09±7.16 | 413.35±12.42 | 403.54±8.28 |
| humanoid venus | 404.27±8.85 | 422.51±10.70 | 399.94±9.23 | 418.17±9.76 | **441.38±8.28** | 403.26±6.79 | 418.94±8.74 | 409.46±7.36 |
| humanoid_standup earth | 15071.09±263.06 | 15326.95±282.80 | 15144.32±357.49 | 15175.28±202.11 | **15840.15±360.31** | 15449.14±340.42 | 15490.03±112.57 | 14912.97±374.86 |
| humanoid_standup neptune | 14750.00±231.55 | **15508.03±196.31** | 15286.00±201.82 | 15241.48±339.27 | 15460.77±310.30 | 14995.58±333.92 | 15362.69±231.12 | 14781.66±189.12 |
| humanoid_standup saturn | 15569.94±424.35 | 15325.96±220.42 | 15523.18±326.58 | 14997.91±201.77 | **15836.01±320.84** | 15606.16±331.10 | 15384.35±301.02 | 15237.92±273.92 |
| humanoid_standup uranus | **15574.27±359.42** | 15556.93±351.34 | 15383.40±180.16 | 15450.26±324.35 | 15559.72±289.16 | 15571.07±336.62 | 15472.19±213.25 | 15097.92±394.43 |
| humanoid_standup venus | 15412.67±338.44 | 14976.94±179.43 | **15625.86±432.62** | 14863.67±218.01 | 15502.47±308.02 | 15260.53±282.17 | 15474.73±230.70 | 14967.62±241.89 |
| inverted_double_pendulum earth | 2317.33±207.61 | 2288.74±184.06 | 2041.60±76.59 | 2406.55±313.44 | **2914.87±625.12** | 2269.29±383.63 | 2018.35±163.15 | 1790.84±109.45 |
| inverted_double_pendulum neptune | 1964.66±131.49 | 2107.74±113.13 | 2013.45±124.46 | 1978.39±142.08 | **2951.25±818.72** | 2253.28±351.84 | 2232.56±266.25 | 1770.85±134.58 |
| inverted_double_pendulum saturn | 2009.62±138.88 | 2150.17±143.98 | 1755.09±193.71 | 2192.51±128.05 | **4091.42±681.94** | 2576.91±334.47 | 2239.90±280.99 | 1827.31±119.52 |
| inverted_double_pendulum uranus | **2662.54±232.24** | 2390.25±173.41 | 2341.58±269.46 | 2168.99±156.42 | 2205.95±252.96 | 2117.50±162.89 | 2257.59±269.15 | 1597.96±170.20 |
| inverted_double_pendulum venus | 2254.82±147.77 | 2271.42±139.16 | 2017.03±71.05 | 2158.85±186.12 | 2199.43±399.36 | 1813.53±105.78 | **2299.24±160.00** | 1740.61±94.51 |
| inverted_pendulum earth | 804.46±36.68 | 862.76±30.91 | 780.84±54.07 | 799.91±46.62 | **902.24±41.24** | 830.42±41.66 | 851.53±39.29 | 704.78±55.58 |
| inverted_pendulum neptune | 809.60±16.01 | 842.74±51.32 | 794.41±41.03 | 731.21±56.04 | **956.44±18.76** | 766.27±64.73 | 837.28±30.11 | 754.82±48.19 |
| inverted_pendulum saturn | 783.85±49.98 | 798.93±39.52 | 830.80±40.41 | 818.36±39.13 | **949.45±19.76** | 829.33±45.46 | 783.08±56.66 | 707.96±51.99 |
| inverted_pendulum uranus | 837.21±30.90 | 817.38±48.22 | 789.76±56.29 | 872.98±17.21 | **958.56±15.93** | 807.50±58.58 | 828.49±67.66 | 710.02±56.69 |
| inverted_pendulum venus | 821.67±40.90 | 829.06±39.68 | 868.14±33.14 | 859.93±44.49 | **909.80±35.26** | 825.11±63.73 | 822.27±42.15 | 715.38±64.15 |
| pusher earth | -1312.38±278.64 | -1112.88±222.44 | -1072.36±161.52 | -1744.33±411.55 | -1201.31±273.90 | -894.35±133.19 | -892.58±121.61 | **-869.27±178.16** |
| pusher neptune | -1080.77±195.92 | -1042.90±195.39 | -1085.76±195.45 | -842.46±119.13 | -1443.96±343.86 | **-775.35±34.41** | -958.06±145.01 | -884.10±114.19 |
| pusher saturn | -1363.24±218.56 | **-787.08±78.07** | -1472.34±252.47 | -1053.19±171.18 | -1792.26±397.92 | -874.96±115.52 | -986.72±140.44 | -1509.28±302.25 |
| pusher uranus | -1024.84±94.57 | -1025.57±200.46 | -1143.05±209.17 | **-835.22±65.40** | -1276.92±185.30 | -964.43±183.80 | -983.23±127.32 | -1226.61±235.12 |
| pusher venus | -1165.77±109.89 | -979.44±181.48 | -1001.89±113.94 | -1786.47±388.82 | -989.84±126.28 | **-969.73±161.02** | -1123.15±220.39 | -1254.35±175.35 |
| reacher earth | -127.13±9.39 | -127.08±8.52 | -116.19±8.80 | -130.28±7.72 | -163.53±9.80 | -132.18±10.82 | -125.36±7.88 | **-69.09±3.10** |
| reacher neptune | -120.52±8.97 | -117.74±10.56 | -127.49±11.35 | -117.46±9.91 | -150.79±6.02 | -120.91±8.81 | -124.09±11.69 | **-75.92±4.27** |
| reacher saturn | -135.61±7.87 | -135.54±7.57 | -139.13±6.37 | -134.98±6.13 | -159.89±5.79 | -128.39±8.88 | -140.82±6.51 | **-72.24±3.49** |
| reacher uranus | -128.50±9.29 | -111.66±12.92 | -127.80±12.80 | -123.17±10.61 | -160.32±5.28 | -106.69±9.92 | -122.99±11.47 | **-67.55±3.83** |
| reacher venus | -128.52±7.52 | -135.53±7.59 | -133.24±7.10 | -129.61±6.66 | -162.77±12.17 | -127.84±10.51 | -125.66±9.49 | **-73.92±4.40** |
| walker2d earth | 533.33±35.11 | 554.08±36.38 | 497.08±23.13 | 560.46±34.99 | 472.29±21.06 | 532.12±31.66 | **564.42±41.73** | 470.66±9.92 |
| walker2d neptune | 477.85±14.71 | 466.67±17.53 | 452.49±24.10 | 468.06±18.19 | 449.31±30.71 | **502.02±36.18** | 495.70±17.19 | 491.05±14.92 |
| walker2d saturn | 544.88±24.64 | 546.21±19.35 | 504.69±12.32 | 513.96±10.34 | 501.25±11.28 | 520.55±24.78 | **578.57±57.13** | 480.43±8.62 |
| walker2d uranus | **603.82±49.26** | 602.40±57.29 | 534.52±19.51 | 579.88±41.28 | 478.63±22.67 | 540.96±39.58 | 580.59±46.77 | 523.32±21.35 |
| walker2d venus | 539.88±27.54 | **588.04±25.65** | 518.03±30.71 | 565.84±23.18 | 476.70±20.70 | 544.60±31.30 | 574.90±29.46 | 502.04±15.44 |

## B.8 Individual Training Plots

In this section, we display the individual training plots for all base environments with the default earth gravity. We show the mean reward and standard error. Table 16 contains the training plots for Brax, and Table 15 contains the plots for classic control.

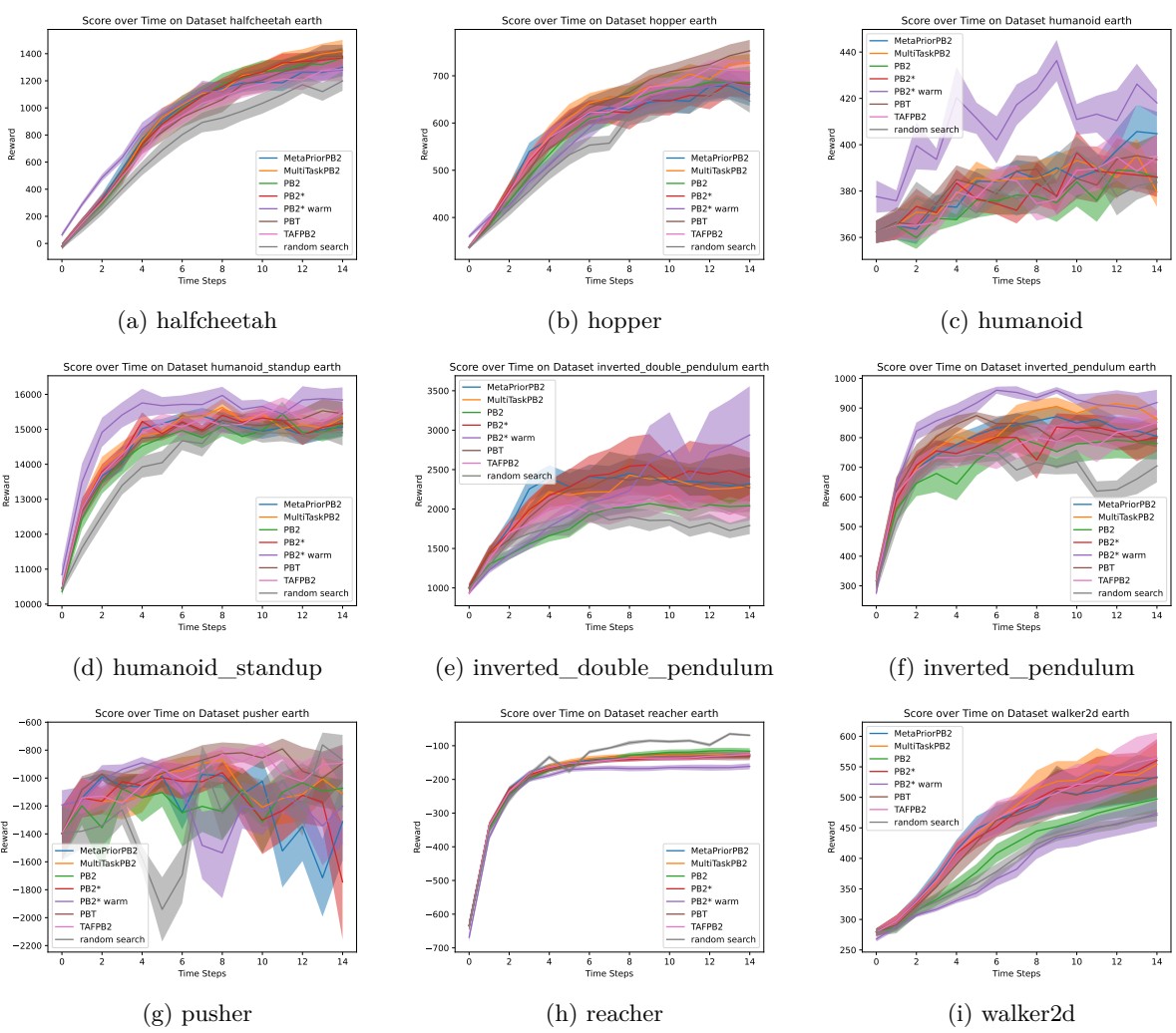

Figure 15: Training plots with earth gravity (default value) for all Brax environments. MultiTaskPB2 is consistently one of the top-performing methods, showing its robustness. It is notable that warmstarted PB2* is the best-performing method on some environments (e.g. *humanoid* and *inverted_pendulum*) while being the worst performing on others (e.g. *reacher* and *walker2d*).

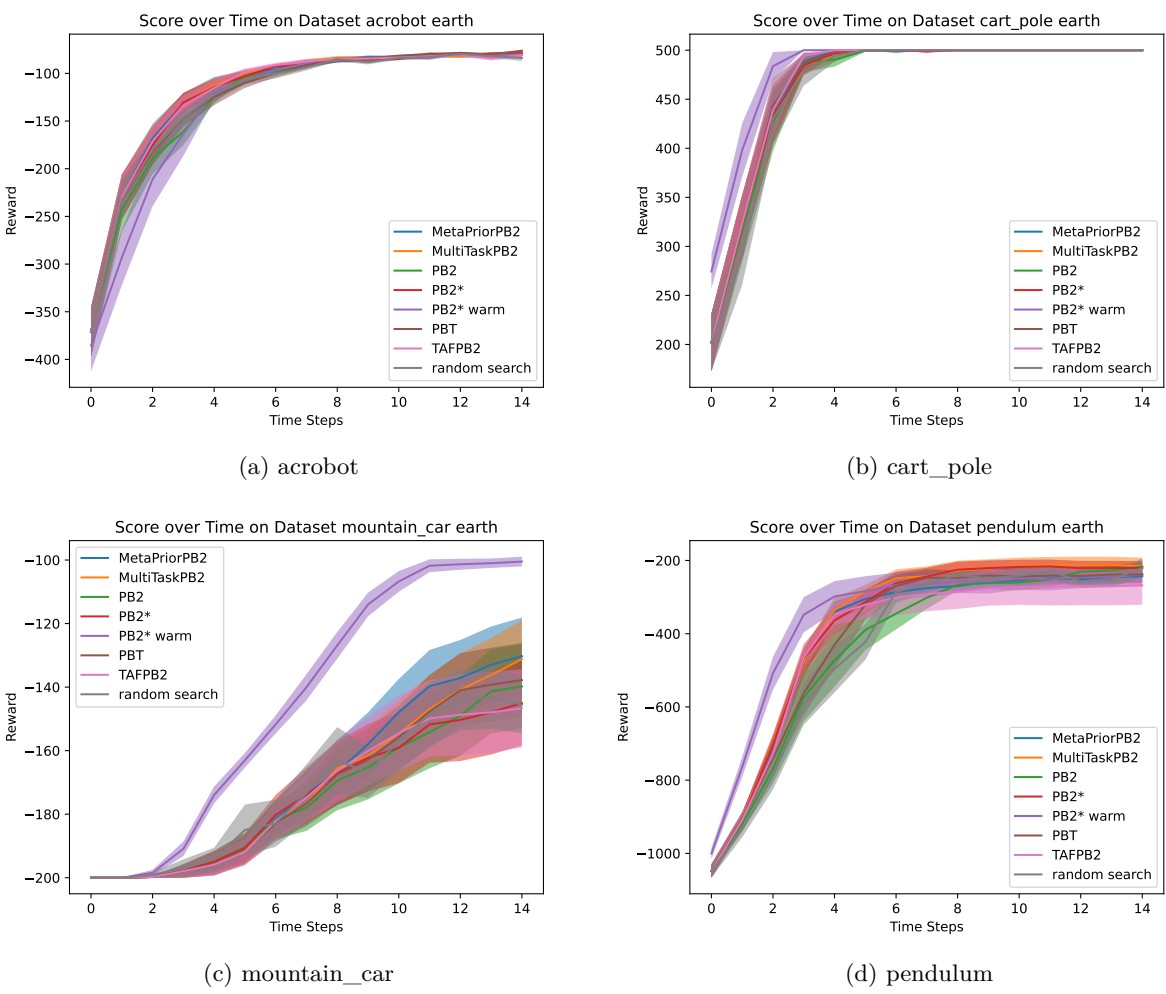

(a) acrobot

(b) cart_pole

(c) mountain_car

(d) pendulum

Figure 16: Training plots with earth gravity (default value) for all classic control environments. Multi-TaskPB2 is consistently one of the top-performing methods, showing its robustness. Warmstarted PB2*, on the other hand, performs inconsistently. In *mountain_car*, it performs exceptionally well, while it performs poorly on *acrobot*.

