# OpenReview forum: "Meta-learning Population-based Methods for Reinforcement Learning"
_TMLR — Accepted by TMLR_

### Review · Reviewer_xanU · 2024-12-12

**Summary Of Contributions:**

This paper proposes improving population based learning using meta-learning methods. In particular, it builds off PB2, also a meta-learning algorithms for Population based training that uses Gaussian Processes. The authors propose 4 enhancements based on using different elements/improvements of GPs. Critically it utilises some meta-data from various environments to help with improving the efficiency.

**Audience:**

Yes

**Broader Impact Concerns:**

No major concerns.

**Claims And Evidence:**

Yes

**Requested Changes:**

No major changes but some clarifications as commented above in the “Questions/Improvement points”.

**Strengths And Weaknesses:**

Strengths:
- Proposed methods demonstrate strong results vs baselines on experimented domains (under certain computational contrains)
- Clear approaches of playing with various levers of BO/GPs to enhance PB2
- Good acknowledgement and awareness of limitations (meta-data alignment and continuous hyperparams, computationally expensive vs baselines)
- Provided open-source code to reproduce experiments

Questions/Improvement Points:
- The contributions section in the introduction should be more detailed and clarified, and summarise your findings/results. For example:  --> “We investigate potential synergies of combining our methods”. What are the synergies? What was the summary/result of this? Good/bad/no-use?
--> Similarly “We investigate the behaviour ……”. What behaviour? Ablation for hyperparamter? So is it sensitive/not sensitive etc.?
- Could you expand why “BO naturally lends itself to meta-learning” in the related work section?
- Why only 15 iterations of the algorithm? Both the baseline algorithms still seem to be improving/have not converged. If there is a particular reason, you should explain why just 15 iterations.

---

> ### Author Response · Authors · 2024-12-25
> **Response to Reviewer xanU**
>
> Thank you for your review. We appreciate your feedback and help in improving the clarity of our work. Below, we will address each of the points you raised:
>
> - We agree that our contributions section could be more specific, and we will revise this section to include more detailed explanations and summaries of our findings.
> - We will expand the reasoning in our paper about why PB2, with its use of BO, is preferable to other population-based approaches. BO builds a model that can be used for meta-learning in various ways. Therefore, BO lends itself more to meta-learning than other model-free approaches.
> - We chose this training length because of the compute costs of our experiments. Since our primary motivation for meta-learning is its initial speed-up, we deemed it acceptable that not all methods converge within the trading budget. However, we agree that it would be of interest to see their behavior for longer training times. We will add an experiment to our paper where we increase the training iterations for all the methods.
>
> We will follow TMLR’s recommendations and wait until all reviews have been submitted before uploading a revised version of our manuscript.
>
> Thank you again for your constructive feedback.

---

### Review · Reviewer_6hor · 2024-12-31

**Summary Of Contributions:**

The authors propose four meta-learning methods tailored for Population-Based Bandits (PB2) to dynamically optimize hyper parameters in RL algorithms. PB2 uses time-varying Gaussian Processes (GPs) and Bayesian Optimization (BO) to make explore-exploit decisions in the hyper parameter search space. But the GP initially lacks sufficient information to make good hyper parameter choices especially in the initial training stages with small number of examples. Instead, the algorithms proposed in this paper introduce different ways to leverage meta-learning over different RL tasks to address this drawback of PB2.
- **TAFPB2**: Enhances the BO acquisition function using meta-learning.
- **MultiTaskPB2**: Employs multi-task Gaussian processes for improved surrogate modeling.
- **MetaPriorPB2**: Constructs a GP prior using meta-learned ensemble.
- **Warmstarting**: Leverages static hyperparameter portfolios to accelerate optimization.

Experiments demonstrate that MultiTaskPB2 outperforms PB2 and other baselines in both anytime and final performance across two RL environment families, classic control and Brax.
The code implementation is shared for reproducibility.

**Audience:**

Yes

**Claims And Evidence:**

No

**Requested Changes:**

Please refer to the weaknesses section.

**Strengths And Weaknesses:**

Strengths:

- Hyperparameter optimization algorithms for RL is an important research question and can improve practical performance of RL algorithms.
- In this paper, authors have clearly described the prior work and how they extend prior frameworks by addressing some of their limitations.
- Ablation experiments have been performed to validate the different components of the proposed algorithms, eg. the weighting mechanism used to combine different meta-tasks, or probability of configuration change at perturbation intervals.


Weaknesses:

- The success of the methods heavily relies on the availability of relevant and well-aligned tasks which can be used for meta-learning. Although the RGPE heat maps seem intuitive, no quantitative metric is provided for task-similarity correctness.

- The overhead introduced by MultiTaskPB2 is significant for smaller environments. The paper minimizes this issue for complex tasks, but practical implications for researchers with limited computational resources should be highlighted more explicitly.

- There is substantial variance in normalized score and rank over iterations, particularly in the classic control environments in Fig 2. This is also true in Fig 4 and Fig 14 for Brax. This makes it difficult to agree to the paper's claims about superior performance of the proposed algorithms.

---

> ### Author Response · Authors · 2025-01-14
> **Response to Reviewer 6hor**
>
> We sincerely thank you for your valuable feedback. Your comments are helping us to improve our work. In the following, we will respond to your concerns.
>
> **Relying on well-aligned tasks and quantitative metric for task-similarity correctness**
>
> Meta-learning approaches, in general, require relevant tasks to learn from. In our case, the set of meta-tasks contained homogeneous environments with the same base environment as the target task but different parameters, i.e., the gravity. We assume this is what you meant by "well-aligned [...] tasks" but we appreciate further clarification.
>
> Reinforcement Learning is badly behaving, which makes it hard to do transfer learning even if the tasks are homogenous [1, 2]. Considering this, demonstrating the effectiveness of our meta-learning approaches on a mixture of homogenous and heterogenous tasks constitutes a significant and relevant contribution.
> If you are aware of any works about successful transfer learning in heterogeneous RL settings, we would be grateful if you could share them with us. Referencing such works would enhance our work by giving it a fuller context.
>
> We agree that there is a lack of quantitative metrics for task-similarity correctness. To address this, we could add a percentage breakdown of the average weight used for tasks with the same base environment, "related" base environments (e.g., humanoid vs. humanoid standup), and "unrelated" base environments in the caption of our RGPE heat maps.
> This would still build on an intuitive understanding of which base environments are related. However, we think this is acceptable, given that defining the similarity of RL environments is still an open research topic.
> We would appreciate your feedback on whether this is sufficient, and we are open to suggestions for other metrics.
>
> **Practical Implications of runtime overhead**
>
> Thank you for pointing this issue out. We agree that we should address the practical implications for researchers with limited computational resources. We will do this in 5.6 RQ5: Runtime Comparison and add another reference in the Conclusion section. Following TMLR’s recommendations, we will wait until all reviews have been submitted before uploading a revised version of our manuscript.
>
> **Variance in normalized score and rank over iterations**
>
> We would appreciate it if you could point out specific performance claims that you find too strong. This would help us adjust our language to better reflect the empirical results.
> We would like to ask if your concern is about the changing of ranks between methods across iterations or the variance of individual methods' performance.
>
> While there is indeed variance in the rankings over iterations in the Figures you mentioned, we observe a consistent trend throughout our experiments where our meta-learning methods outperform the baselines. Additionally, the critical difference diagram for anytime performance takes the variance into account and aligns with our conclusions.
>
> In Figure 4, the variance between the different methods can partly be attributed to their distinct optimization focuses. Warmstarting is designed to provide early-iteration benefits, which leads to rank changes as training progresses.
> This behavior is fundamentally different from static warmstarting approaches and explains some of the ranking variations observed.
>
>
> **References**
>
> [1] Kirk, Robert, et al. "A survey of zero-shot generalisation in deep reinforcement learning." Journal of Artificial Intelligence Research 76 (2023): 201-264.
>
> [2] Eimer, Theresa, Marius Lindauer, and Roberta Raileanu. "Hyperparameters in reinforcement learning and how to tune them." International Conference on Machine Learning. PMLR, 2023.

---

> > ### Comment · Reviewer_6hor · 2025-02-26
> > **Thank you for the response**
> >
> > Thank you for addressing the requested changes.
> >
> > > If you are aware of any works about successful transfer learning in heterogeneous RL settings, we would be grateful if you could share them with us. Referencing such works would enhance our work by giving it a fuller context.
> >
> > [1] includes some empirical results using meta-learning approaches for adapting across bandit agents with different exploration algorithms, although the problem setting is different than this work.
> >
> >
> > Regarding variance in the rankings, I was mostly concerned about the lack of distinction in performance across the different proposed methods. I have read the author's response to reviewer bF8L's comment about similar observations.
> >
> > [1] Banerjee et al. 2023, "MERMAIDE: Learning to Align Learners using Model-Based Meta-Learning" TMLR.

---

> > > ### Author Response · Authors · 2025-03-10
> > > **We included the reference**
> > >
> > > Thank you for your suggestion. To better contextualize our work, we've included Banerjee et al.'s work in our related work on meta-learning.

---

### Review · Reviewer_bF8L · 2025-01-28

**Summary Of Contributions:**

The paper proposes a few ways to incorporate prior information into population based training or population based bandits, where hyper-parameters of the training system are optimized over the course of training. Compared to baseline methods, they propose ways to incorporate prior information that helps avoid the initial warmup slowness and accelerate learning. They show improvements in a number of benchmarks.

**Audience:**

Yes

**Claims And Evidence:**

No

**Requested Changes:**

# ==== figure 2 results ====

In Figure 2, the baseline methods (random and PB2) do not seem to have significant score improvements over the course of training. In this case, the random search does not seem to improve over iterations at all while the PB2 baseline seems to improve a little bit during the whole process, with an initial dive in performance. How should we interpret the dive in performance at the beginning of PB2, is this related to the warm-up motivation?

If the random parameter search does not improve the new iterates' performance, it should still improve the cumulative performance, that is the best hyper-parameter config located by different algorithms thus far in the iterations. Does that sound reasonable?

Though the paper claims that MultiTaskPB2 performs the best, it is still not very clear if it is indeed the statistically significantly the best approach among the four proposed new methods. The error bars from different techniques rather overlap with each other quite a lot, making the statement less clear.

# === Fig 14 ===

It feels like Fig 14 paints a more comprehensive comparison across different methods, where it seems that even random search obtains reasonable performance in a few environments such as pusher and reacher, where random search performs the best. What is the special property of such environment that makes random search stand out in such a case, where deliberate hyper-parameter search techniques do not well as well?

# ==== nit comments ====

Figure 6 runtime comparison can be more meaningful if put in relative values rather than absolute values.

Across Fig 4-5 it feels like the ranking curves are a bit redundant given the normalized scores, and the results might be more presented using tables rather than plots.

**Strengths And Weaknesses:**

The paper studies an interesting problem of incorporating prior information into population based bandit problems. The paper also offers a few rather intuitively sensible methods. The technical contributions are of interest to the general ML community and especially interesting for population based training practitioners.

The main weakness might lie in that it's a bit hard to tell apart the relative strengths of the proposed four methods. The paper rather feels like a bag of tricks that can accelerate population based training with hyper-parameter optimization. The connection between different methods feels rather less coherent, and the paper feels like just enumerating different possibilities without stressing enough on the common insights.

---

> ### Author Response · Authors · 2025-02-11
> **Response to Reviewer bF8L**
>
> Thank you for your thoughtful feedback and recommendations to improve the quality of our paper. In the following, we will address the mentioned weaknesses and requested changes.
>
>
> **Weaknesses - Contrasting the Methods**
> Our motivation for presenting multiple methods is that meta-learning is not yet applied in the time-varying setting of dynamic HPO.  Therefore, we try to cover the vast types of meta-learning methods with representatives to identify promising research directions.
> We achieve the common insight that meta-learning is useful in this setting (Section 5.2) and see the contrast of the different methods throughout the paper.
> For example, we observe that  MetaPriorPB2, MultiTaskPB2, and TAFPB2, which all need to determine a task similarity based on observed hyperparameter performance, have less initial performance increase than our warmstarting method (Section 5.2).  We see that MetaPriorPB2 suffers if the hyperparameter configurations evaluated in the target task do not match the configurations in the meta data because the prior can only inform in these regions (Section 5.4).  We identify that MultiTaskPB2, compared to other methods, avoids bad hyperparameters that introduce numerical instability into the training (Section B.6).
>
> If you think it would help with clarity, we could stress these insights more or add a subsection at the end of the experiments section to summarize the similarities and differences of the different meta methods.
>
> **Figure 2 - Performance Decrease During Training**
> The question about performance decline appears to misunderstand Figure 2's normalized scoring. All plots are relative, and the normalized scores are normalized considering the performances after each iteration, not the entirety of the training. Therefore, the graphs tell us that random search does not improve its relative performance compared to the other methods but does not say anything about improved overall score. Additionally, we want to emphasise that random search is not dynamic and does not sample new configurations at each iteration; it simply trains its initially sampled configurations in parallel.
>
> Yes, the initial dive in PB2's relative performance is related to our meta-learning motivation. PB2 chooses which hyperparameters to try next using the observed performance of the hyperparameter configurations it trained. At the beginning of training, only the performance of a few configurations is available, which makes for a relatively uninformed selection of new hyperparameters. We use meta-data to inform this decision and remedy the initial lack of information.
> We can also observe that warmstarting initially outperforms the other methods using meta-learning. This is because they need to determine task similarity based on the performance of hyperparameter configurations, which necessitates some evaluations, and our warmstarting method uses a portfolio of configurations with good expected initial performance.
>
> **Figure 2 - Statistical Significance of MultiTaskPB2's Performance**
> To test for statistical significance of the anytime performance of different methods, we include a critical difference diagram \[1\] at the right side of our main visualizations. A critical difference diagram visualizes the results of a Friedman test followed by a Nemenyi post-hoc test, which tests if there are statistically significant differences between the mean ranks of a set of predictors.
> In Figure 2, we can see that MultiTaskPB2 statistically significantly outperforms all other methods except MetaPriorPB2 on Brax and warmstarted PB2\* on classic control.
> It still outperforms both methods, but the difference is not statistically significant.
> This result, in combination with the robustness that MultiTaskPB2 demonstrates in our ablation studies, justifies our performance claims of MultiTaskPB2.
>
>
> **Figure 14 - Surprisingly Good Random Search Performance**
> This phenomenon has been observed in other studies \[2,3\], where random search serves as a well-performing baseline, particularly under low budget constraints \[3\]. The reasons and conditions for this solid performance in the RL setting are not yet fully understood. We agree that conducting a broader study on this topic would be interesting and potentially insightful but outside of the scope of this paper.
>
> **Figure 6 - Relative Values for Runtime Comparison**
> The plots in Figure 6 use a double y-axis; one is absolute, and the other is relative to our PB2 benchmark. We think showing both axes presents a fuller picture of the runtime constraints.

---

> > ### Author Response · Authors · 2025-02-11
> > **Response to Reviewer bF8L Part 2**
> >
> > **References**
> >
> > \[1\] Demšar, Janez. "Statistical comparisons of classifiers over multiple data sets." _The Journal of Machine learning research_ 7 (2006): 1-30.
> >
> > \[2\] Shala, Gresa, et al. "HPO-RL-Bench: A zero-cost benchmark for HPO in reinforcement learning." _AutoML Conference 2024 (ABCD Track)_. 2024.
> >
> > \[3\] Eimer, Theresa, Marius Lindauer, and Roberta Raileanu. "Hyperparameters in reinforcement learning and how to tune them." _International Conference on Machine Learning_. PMLR, 2023.

---

### Author Response · Authors · 2025-03-10
**Notification of Revised Manuscript Submission**

Dear Reviewers, We have uploaded a revised version of the manuscript, incorporating your valuable suggestions. The modifications are highlighted in red for your convenience. Additionally, please note that there are changes in the appendix as well. We hope these updates address all the remaining concerns raised in your reviews.

---

### Decision · Action_Editor_XDdn · 2025-03-05

**Recommendation:** Accept as is

**Comment:**

The reviewers were mostly positive about the paper in their initial reviews and they acknowledged the rebuttal has resolved most of the raised concerns. Thus, I am proposing to accept the paper as is. However, if the authors are interested in improving their manuscript for the camera ready version, reviewers' suggestions include:
- Providing "a more convincing argument in support of the variance in the plots which makes it hard to distinguish among the different proposed methods";
- "The empirical setups can be made more solid, such as the ablation against why random search turns out to well perform in certain cases, and justifies whether more sophisticated approaches are needed at all".

Although not all of the reviewers provided a positive final recommendation, I think this can be a valuable contribution to the TMLR community. A wide adoption of the proposed method may instead be hindered by a significant computational burden to integrate it, which the authors acknowledged in their conclusions.

**Audience:**

This paper addresses the problem of exploiting previous runs to warm-start a popular method for online hyperparameter optimization. It can be interesting for the RL community and practitioners.

**Claims And Evidence:**

The reviewers agree that the claims made in the paper are mostly supported by the provided evidence,